# Whole cell reconstructions of *Leishmania mexicana* through the cell cycle

**Molly Hair[1], Ryuji Yanase[1], Flávia Moreira-Leite[1], Richard John Wheeler[2], Jovana Sádlová[3], Petr Volf[3], Sue Vaughan[1]\*, Jack Daniel Sunter[1]\***

**1** Department of Biological and Medical Sciences, Oxford Brookes University, Oxford, United Kingdom,
**2** Peter Medawar Building for Pathogen Research, University of Oxford, Oxford, United Kingdom,
**3** Department of Parasitology, Charles University, Prague, Czech Republic

\* svaughan@brookes.ac.uk (SV); jsunter@brookes.ac.uk (JDS)

**Data Availability Statement:** The SBF-SEM dataset generated and analysed for this study have been deposited and is available in the EMPIAR repository database [54] under the EMPIAR ID: 11826 -

## Abstract

The unicellular parasite *Leishmania* has a precisely defined cell architecture that is inherited by each subsequent generation, requiring a highly coordinated pattern of duplication and segregation of organelles and cytoskeletal structures. A framework of nuclear division and morphological changes is known from light microscopy, yet this has limited resolution and the intrinsic organisation of organelles within the cell body and their manner of duplication and inheritance is unknown. Using volume electron microscopy approaches, we have produced three-dimensional reconstructions of different promastigote cell cycle stages to give a spatial and quantitative overview of organelle positioning, division and inheritance. The first morphological indications seen in our dataset that a new cell cycle had begun were the assembly of a new flagellum, the duplication of the contractile vacuole and the increase in volume of the nucleus and kinetoplast. We showed that the progression of the cytokinesis furrow created a specific pattern of membrane indentations, while our analysis of sub-pellicular microtubule organisation indicated that there is likely a preferred site of new microtubule insertion. The daughter cells retained these indentations in their cell body for a period post-abscission. By comparing cultured and sand fly derived promastigotes, we found an increase in the number and overall volume of lipid droplets in the promastigotes from the sand fly, reflecting a change in their metabolism to ensure transmissibility to the mammalian host. Our insights into the cell cycle mechanics of *Leishmania* will support future molecular cell biology analyses of these parasites.

## Author summary

The parasite *Leishmania* causes the insect-transmitted neglected tropical disease, leishmaniasis. *Leishmania* is a single-celled parasite with a distinctive and highly defined shape. It is critical that each generation of the parasite retains the same shape. To achieve this, the parasite has a highly coordinated cell cycle in which organelles and key structures are duplicated and segregated at specific points. Previous work has defined a framework of cellular changes at the light microscopy level but this has limited resolution. We used volume electron microscopy to reconstruct *Leishmania* cells at different points in the cell

https://www.ebi.ac.uk/empiar/EMPIAR-11826/. The electron tomography dataset generated and analysed for this study has been deposited and is available in the Zenodo repository - https://zenodo.org/records/10652021.

**Funding:** MH was funded by a Nigel Groome studentship at Oxford Brookes University. This work is supported by the Wellcome Trust to JDS (221944/Z20/Z) and to RJW (211075/Z/18/Z and 103261/Z/13/Z). RY was supported by a JSPS Overseas Research Fellowship. PV was supported by the Czech Science Foundation (GACR 21-15700S). The funders had no role in study design, data collection and analysis, decision to publish, or preparation of the manuscript.

cycle, which provided a three-dimensional overview of organelle positioning, duplication, and segregation. Moreover, we found that cytokinesis created membrane indentations that persisted in the daughter cells for a limited time. Finally, we examined *Leishmania* parasites in their insect vector, the sand fly. We found that the parasites in the sand fly had a greater number of lipid droplets, indicative of changes to their metabolism. Our insights into the cell cycle mechanics of *Leishmania* will provide a framework for future analyses of these parasites.

## Introduction

*Leishmania* spp. are flagellated unicellular parasites that cause the neglected tropical disease, leishmaniasis. *Leishmania* are transmitted between mammalian hosts by its sand fly vector and during its life cycle the parasite has two developmental stages each principally defined by the shape and form of the cell and flagellum length, with limited molecular makers to differentiate the promastigote forms [1,2]. In the sand fly midgut, *Leishmania* has a promastigote morphology with an elongated cell body and a single flagellum extending from the anterior end of the cell body. Promastigote cells have a highly polarised layout with single-copy organelles occupying defined positions within the cell. The nucleus is positioned centrally within the cell body, with the concatenated mitochondrial DNA (kinetoplast) found towards the anterior end of the cell. The kinetoplast is physically connected to the basal body from which the flagellum extends. The base of the flagellum is within a membrane invagination called the flagellar pocket, which is the sole site of endo/exocytosis. In the mammalian host, *Leishmania* resides within the parasitophorous vacuole of a macrophage as an amastigote. This form is characterised by having a smaller and more spherical cell body, with a flagellum that just extends beyond the exit of the flagellar pocket [3–6].

*Leishmania* are members of the trypanosomatid parasites that includes the other human pathogens *Trypanosoma brucei* and *Trypanosoma cruzi*. Life cycle stages of these parasites have highly defined morphologies, with two major morphological supergroups–liberform and juxtaform. *Leishmania* belong to the liberform group, with *T. brucei* and *T. cruzi* in the juxtaform group [7]. The defining feature of these different morphological groupings is the length of the portion of the flagellum that is laterally attached to the cell body. In liberform *Leishmania*, there is a minimal lateral attachment of the flagellum to the cell body within the flagellar pocket [8], whereas in juxtaforms such as *T. brucei* there is an extended region of flagellum attachment along the majority of the cell body [9]. This lateral attachment in *T. brucei* and *L. mexicana* is mediated by a complex cytoskeletal structure called the flagellum attachment zone (FAZ). In *T. brucei* the FAZ is important for regulating cell division through defining the start point of cytokinesis furrow ingression [9,10] whilst in *Leishmania*, the FAZ is important for defining flagellar pocket shape and function [6]. The distinctive cell morphology of a *Leishmania* promastigote is retained in each subsequent cell generation, requiring cell division to occur reproducibly with high-fidelity [4,7]. Despite the liberform likely being the ancestral morphology of the trypanosomatid parasites, cell division has been more extensively studied in the juxtaform *T. brucei and T. cruzi* [7,11–13]. However, the timings of and the morphological changes during the *Leishmania* promastigote cell cycle have been examined at the light microscopy level, using a combination of markers for cellular structures (nucleus, kinetoplast, spindle, flagellar pocket), DNA replication, and overall cell morphology [3,4,14–16]. For the vast majority of the cell cycle the parasite has one kinetoplast, one nucleus and one flagellum protruding from the flagellar pocket. During the promastigote cell cycle, the *Leishmania* cell

body undergoes dramatic changes in its length, having doubled in length during G1 then plateauing in S phase before rapidly becoming shorter and wider as the cell divides [3,4]. Nuclear and kinetoplast S phase are near synchronous and around the end of S phase a new flagellum becomes visible protruding from the flagellar pocket alongside the old flagellum. Though given the ~2 μm depth of the flagellar pocket [8], the exact point in the cell cycle at which new flagellum growth starts is unknown. The kinetoplast and nucleus divide at similar times as the cell body shortens, with the flagellar pocket segregating around this point, before the formation of a cytokinesis furrow with cytokinesis proceeding along the long axis of the cell body from the anterior to posterior [3,4].

These light microscopy studies provided an overall temporal and morphological framework for the *Leishmania* cell cycle. This is, however, limited in detail to large organelles—although there are isolated exceptions using specific molecular markers, such as used to analyse the cell cycle related dynamics of the multivesicular tubule lysosome [17]. Our understanding of the timing and manner of cell morphogenesis and organelle inheritance in *Leishmania* is limited. In a previous study, our group used a volume electron microscopy (vEM) approach called serial block face scanning electron microscopy (SBF-SEM)—which is based on automated and sequential sectioning and imaging by SEM—to produce a three-dimensional spatiotemporal map of cell division in *T. brucei*, with ultrastructural detail [18]. This work demonstrated the power of complementary microscopy methods to understand the mechanisms that control trypanosome and by extension juxtaform cell morphogenesis.

Here, we used the vEM methodologies of SBF-SEM and serial electron tomography to examine *Leishmania mexicana* promastigotes at many stages of the cell cycle in unprecedented detail. This provides a comprehensive spatial overview of the changes that occur in organelle shape, number, size, and position through the cell cycle, along with changes in the organisation of the sub-pellicular microtubule cortex, which defines cell shape, and differences between axenically cultured and sand fly inhabiting promastigotes. In doing so, we have provided important insights into the cell cycle mechanics of these parasites and the liberform morphological supergroup.

## Results

### SBF-SEM allows classification of axenic promastigotes into seven characteristic cell cycle stages

Promastigote form *Leishmania mexicana* cells from an asynchronous exponential phase *in vitro* culture were imaged by SBF-SEM, generating a dataset containing 750 individual slices (Z-slice = 100 nm) (Movie 1). We analysed a total of 41 promastigote cells whose cell body and flagella lay within the imaged dataset, representing seven different stages of the cell cycle (see below). The cell body membrane, flagellum, nucleus, mitochondrion with kinetoplast, Golgi, contractile vacuole, paraflagellar rod, lipid bodies, acidocalcisomes and glycosomes were segmented for each individual whole cell (Fig 1A and 1B, S1 and S2 Movies). Organelles were identified as per previous descriptions in transmission electron microscopy [14,19,20]. The volume and surface area was calculated from the segmentation of each cellular structure and organelle and counts were made of multi-copy organelles, shown below and in S1 Table. The endoplasmic reticulum (ER) was segmented in each slice of the dataset to illustrate positioning within each cell, but due to its undulating nature these fragments are not connected across sections to produce a 3D model due to the limitations of Z-resolution. The sub-pellicular microtubules were also not modelled as they are separated by only 20 nm thus were only visible when orientated parallel to the Z axis.

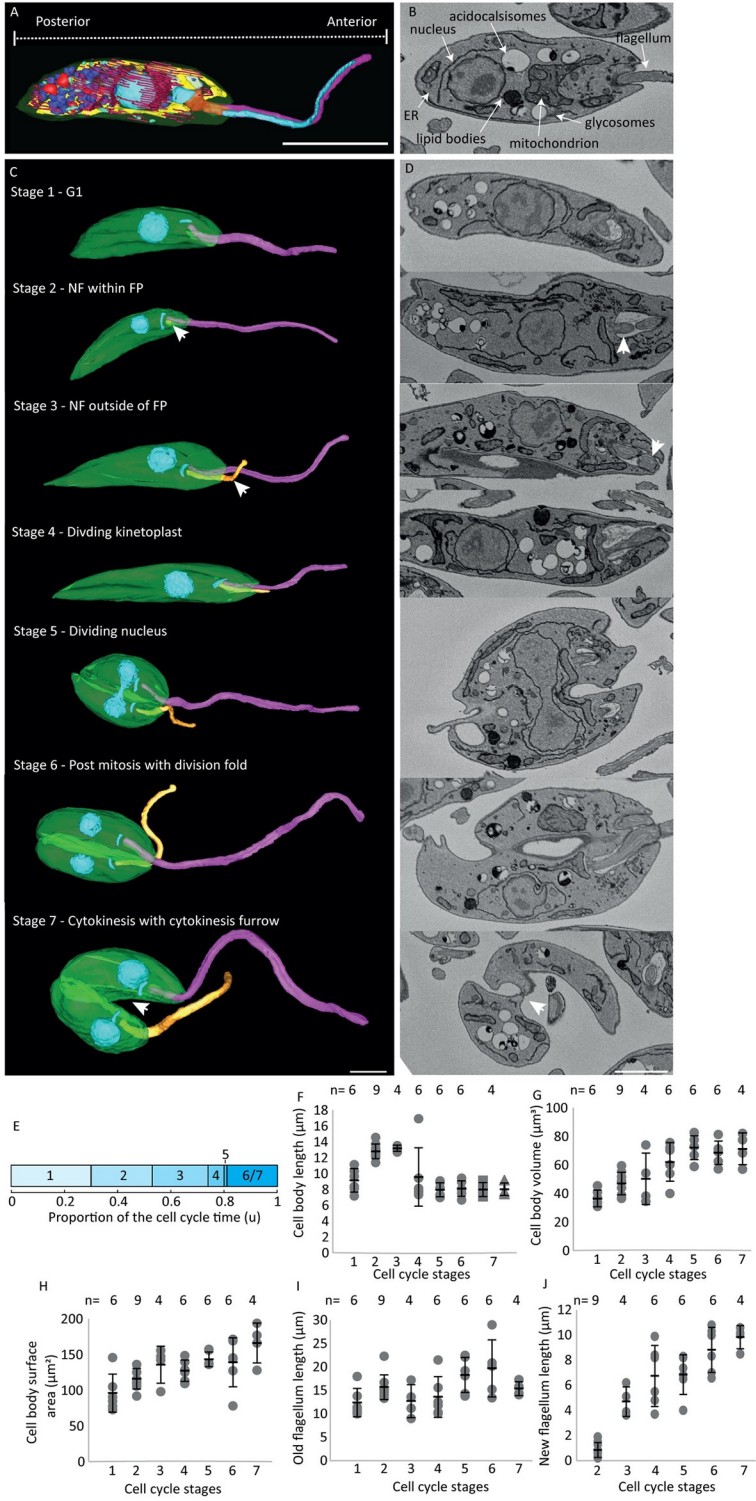

**Fig 1. Distinct ultrastructure features were used to define seven cell cycle stages.** (A) 3D reconstructed Stage 1 promastigote cell showing the spatial organisation of organelles within the cell body. Organelles are modelled using the following colours: cell body (opaque green), flagellum (pink), nucleus (blue), kinetoplast (blue), mitochondrion (yellow), acidocalcisomes (dark purple), lipid bodies (red), glycosomes (pale green), contractile vacuole (dark blue), Golgi (purple), ER (pink), PFR (teal), flagellar pocket (deep red). Scale bar = 5 μm. (B) SBF-SEM data slice of a Stage 1 promastigote cell highlighting the ultrastructural features used to identify organelles. Scale bar = 2 μm. (C-D) 3D

reconstruction and SBF-SEM data slice of seven distinct cell cycle stages representing the entire promastigote cell cycle. Organelles are modelled using the following colours: cell body (opaque green), old flagellum (pink), new flagellum (orange), nucleus (blue), kinetoplast (blue). (Stage 1) Cell with a single flagellum, kinetoplast and nucleus. (Stage 2) New flagellum (white arrow) wholly within the flagellar pocket. (Stage 3) Cell with a new flagellum that has emerged from the anterior cell tip (white arrow). (Stage 4) Cell with an elongated (dividing) kinetoplast and a single nucleus. (Stage 5) Cell with 2 kinetoplasts and a dividing nucleus. (Stage 6) Cell with two separate kinetoplasts and nuclei, with a division fold forming. (Stage 7) Cells with cytokinesis furrow ingression progressing from the anterior end of the cell (white arrow). Scale bar = 2 μm. (E) The approximate timings using ergodic principles of the seven cell cycle stages during the cell cycle. n = 500. (F) Dot plot showing cell body length across the seven cell cycle stages defined. Stage 7 was split into two categories: nascent daughter cell inheriting the old flagellum (square data points) and nascent daughter cell inheriting the new flagellum (triangle data points). Dot plot showing (G) cell body volume, (H) cell body surface area, (I) old flagellum length and (J) new flagellum length across the seven cell cycle stages defined. Error bars show ±standard deviation. Total number of cells per stage: 6 cells in Stage 1, 9 cells in Stage 2, 4 cells in Stage 3, 6 cells in Stage 4, 6 cells in Stage 5, 6 cells in Stage 6 and 4 cells in Stage 7.

Using the asynchronous exponential phase culture means the cells are randomly sampled from all stages of the cell cycle [11,21]. We defined seven distinct cell cycle stages by identifying the number of kinetoplasts and nuclei and the length of the new flagellum, which will enable us to examine changes in both cell and organelle morphogenesis through the cell cycle (Fig 1C and 1D). Stage 1–3 have a single kinetoplast and nucleus and are distinguished by the flagella: Stage 1) Single flagellum (S1 Movie); Stage 2) Two flagella, with the new flagellum (NF) restricted to the flagellar pocket (Fig 1C and 1D; arrow—NF); Stage 3) New flagellum that has emerged from the flagellar pocket (Fig 1C and 1D; arrow—NF); Stage 4–7 are distinguished by kinetoplast, nucleus and cell body division stage: Stage 4) Single elongated (dividing) kinetoplast and a single nucleus (Fig 1C and 1D; arrow–kinetoplast); Stage 5) Two kinetoplasts and a dividing nucleus (Fig 1C and 1D; arrow–nucleus); Stage 6) Two kinetoplasts and two nuclei, with a cell body division fold (Movie 2); Stage 7) Cell body cytokinesis furrow from the anterior end of the cell towards the posterior end (Fig 1C and 1D; arrow—furrow).

For a population in exponential growth it is possible to calculate time spent in each cell cycle stage, assuming that all cells are undergoing the same replicative cycle (Fig 1E) [21]. Here, we represent timings as proportion of time (in units, u) through the ~8 h cell cycle [4]. The new flagellum begins to grow within the flagellar pocket in Stage 2 at 0.3 u and emerges from the flagellar pocket in Stage 3 at 0.53 u. The kinetoplast shows a more elongated form, indicative of the earliest stages of reorganisation for division (Stage 4), at 0.76 u followed by a late mitotic nucleus (Stage 5) characterised by two distinct nuclear lobes linked by a cytoplasmic mitotic nuclear bridge at 0.8 u with segregation of the kinetoplast and nucleus (Stage 6) occurring at 0.81 u.

One of the most notable morphological features of *Leishmania* is the change in cell body length during the cell cycle (Fig 1F). The overall change in cell length agrees with our previous light microscopy analysis [4]. Despite a reduction in cell body length, the cell volume increased from Stages 1–5 and then remained relatively constant through Stages 6–7 (Fig 1G). The universal presence of sub-pellicular microtubules at even spacing means cell surface area is proportional to the total amount of microtubules/tubulin needed for the sub-pellicular array. The cell surface area increased steadily across the cell cycle and therefore so did the amount of tubulin in the sub-pellicular array (Fig 1H). This shows that the cell volume and surface area increases whilst the cell body length decreases as the cell undergoes large cellular rearrangements in preparation for cytokinesis.

## New flagellum does not rotate around the old flagellum

In *L. mexicana* promastigotes the single flagellum remains assembled throughout the cell cycle with the new flagellum growing alongside, as is typical of trypanosomatid parasites [4,7].

Consistent with previous light microscopy [4], the old flagellum length was variable but was always longer than the new flagellum (Fig 1I). By the end of the cell cycle, the new flagellum length (Fig 1J) did not reach that of the old, with flagellum growth therefore continuing after cytokinesis.

In *T. brucei* trypomastigotes the new flagellum extends into the existing flagellar pocket on the anterior side of the existing flagellum. The basal body and new flagellum then rotate around the old flagellum to position the new flagellum towards the posterior of the cell. This rotation occurs prior to the new flagellum extending out of the flagellar pocket and flagellar pocket duplication [22]. Here, we investigated if this rotation occurs during new flagellum growth in *L. mexicana*. The axoneme, paraflagellar rod and flagellar pocket were segmented in cells (Fig 2A). The paraflagellar rod is always positioned alongside microtubule doublets 4–7 of the axoneme [23] and the paraflagellar rod is present on axonemes within the flagellar pocket neck of *L. mexicana* [6]. Therefore, we used this feature as a positional marker. The new flagellum was always (n = 25) positioned next to the side of the old flagellum with the PFR (Stage 2 —Fig 2B; Stage 3 –Fig 2C) indicating that new flagellum rotation does not occur in *Leishmania* and suggests that this phenomenon is restricted to parasites with a trypomastigote morphology [7,22]. This raised a further question regarding the timing of flagellar pocket duplication relative to the elongation of the new flagellum. The new flagellum extended into the existing flagellar pocket bulbous domain in Stage 2 (Fig 2B). In Stage 3, the new flagellum exited the existing flagellar pocket, but flagellar pocket duplication had not occurred (Fig 2C). The first morphological indicator of flagellar pocket duplication was seen in Stage 4 cells, with a ridge of cytoplasm invading the bulbous domain of the existing flagellar pocket between the old and new flagellum (Fig 2D; arrow). This cytoplasmic ridge has previously been observed in *T. brucei* flagellar pocket duplication [22]. Prior to the ridge forming, the old and the new flagellum share the same flagellar pocket exit point (Fig 2E; Stage 2–3). Whilst the ridge is forming from the base of the flagellar pocket, the exit point of the flagellar pocket was still shared by the two flagella (Fig 2E; Stage 4). Two separate exit points for each flagellum were first observed in Stage 5 cells, indicating completion of flagellar pocket duplication by this cell cycle stage (Fig 2E; Stage 5). This sharing and separation between the new and old flagellum exit points can also be seen by whole mount negative staining of the cytoskeleton of dividing cells (Fig 2F and 2G). This shows that flagellar pocket duplication occurs in two stages with the division of the bulbous domain by the cytoplasmic ridge followed by the division of the flagellar pocket neck (Fig 2D) [24].

The flagellum is assembled at its distal tip, with material transported to the site of assembly by the intraflagellar transport system [25,26]. Regulatory mechanisms controlling, for example axoneme stability, are therefore likely to operate at the distal tip of the flagellum [27]. We observed an elaborate cap structure at the flagellum distal tip that has previously been identified in other trypanosomatid parasites including *L. major* (Fig 2H–2J) [28]. The cap structure consisted of two annuli, one associated with the central pair that sits inside the outer doublets that are capped by a second annulus. The annulus associated with the central pair is readily observable in whole mount cytoskeletons (Fig 2J). This elaborate cap structure is only present on the old flagellum and is therefore added post-assembly, with a potential role in beat regulation and axoneme stability [27–29].

## Contractile vacuole and Golgi re-position during the cell cycle

Our data confirmed there was a single contractile vacuole and Golgi located close to the flagellar pocket in Stage 1 cells (Fig 3A), supporting our previous work [8]. During the cell cycle, one additional contractile vacuole and Golgi were observed. A second contractile vacuole was

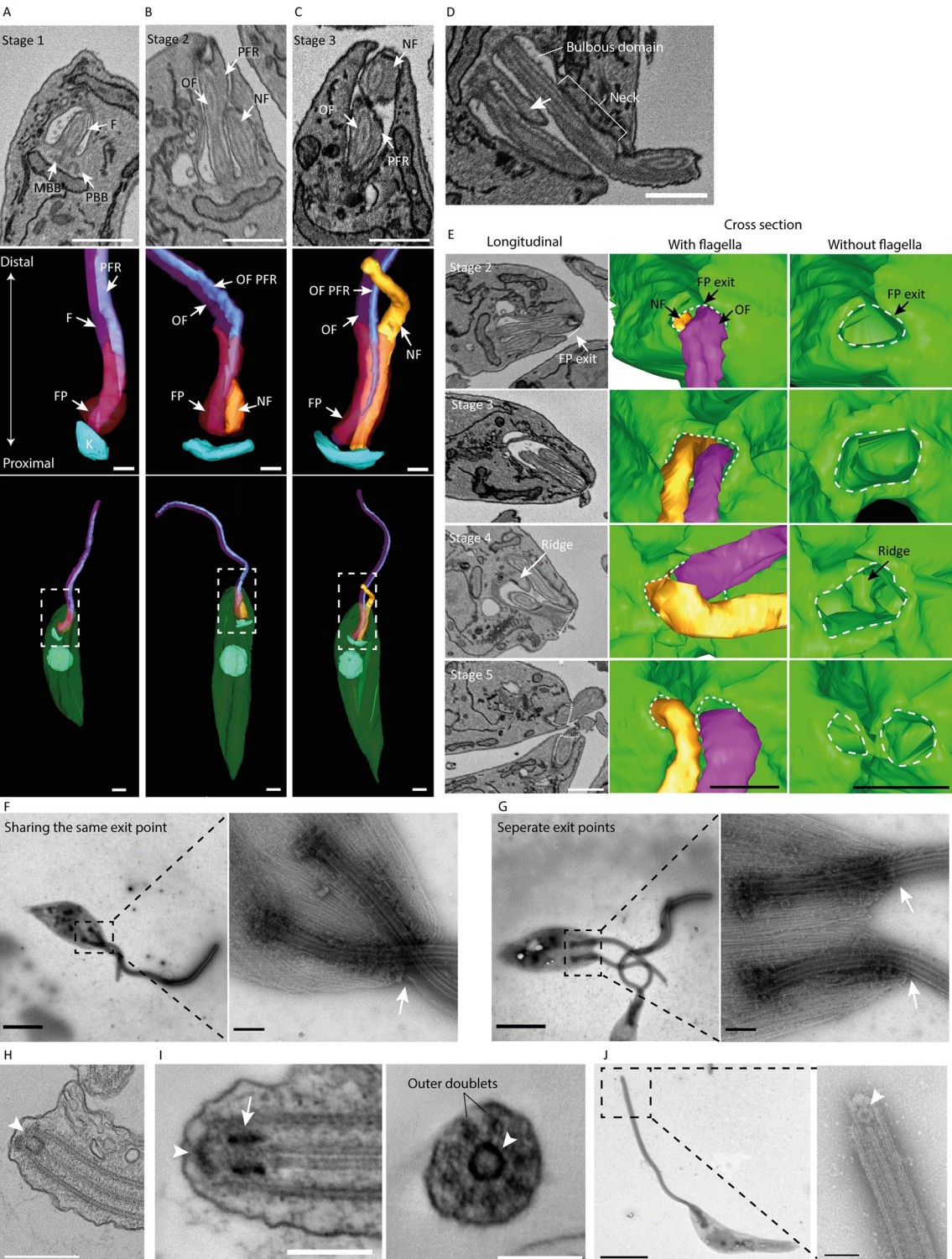

**Fig 2. New flagellum rotation around the old flagellum does not occur.** (A-C) SBF-SEM data slice (scale bar = 1 μm) and two 3D reconstructed models (scale bar = 2 μm) of a Stage 1, 2 and 3 cell illustrating that there is no rotation of the new flagellum (orange) around the old flagellum (opaque pink) during new flagellum growth within and extending out of the flagellar pocket (FP—opaque red) The old flagellum paraflagellar rod (PFR–dark blue) is used as an orientation marker demonstrating the new flagellum is always on the right-hand side of the old flagellum PFR. Mature basal body (MBB), pro basal body (PBB), kinetoplast (K–blue). (D) SBF-SEM data slice showing a cytoplasmic ridge (arrow) within an existing flagellar pocket. Scale bar = 1 μm. (E) Longitudinal SBF-SEM data slice of

Stage 2–5 cells showing the flagellar pocket exit (white dashed line). Cross sections of 3D reconstructed Stage 2–5 cell illustrating that the old flagellum (pink) and new flagellum (orange) share the same flagellar pocket (FP) exit (white dashed line) until Stage 5 post cytoplasmic ridge has formed and two separate flagellar pocket exits are present indicating flagellar pocket duplication is complete. Stage 4 shows the positioning of the cytoplasmic ridge (arrow) within the existing flagellar pocket, but the two flagella still share the same exit point. Scale bar = 1 μm. (F) Whole cell negative stain mounts of a cell with two flagella sharing the same flagellar pocket exit point (arrow). Scale bar = 10 μm. (G) Whole cell negative stain mounts of a cell with two flagella with two separate flagellar pocket exit points (arrow). Scale bar = 1 μm. (H) Transmission electron image of a flagellum with a cap at the distal tip (arrowhead). Scale bar = 500 nm. (I) Serial electron tomogram of the *L. mexicana* flagellar tip in longitudinal and cross-section. There is one annulus that sits within the outer doublets (arrow) and a second annulus that caps the outer doublets (arrowhead). (J) Whole mount detergent extracted *L. mexicana* cell. Higher magnification micrographs of the flagellum distal tip showing the capping structure (arrowhead). Scale bar = 5 μm. Inset scale bar = 200 nm.

first visible in Stage 2 and a second Golgi in Stage 3. (Fig 3A). The appearance of the second contractile vacuole along with the new flagellum are therefore the earliest ultrastructural changes we identified. The second contractile vacuole and Golgi initially appeared alongside the existing organelles (Fig 3B). However, during Stage 4, a single contractile vacuole and Golgi were located next to each separate flagellar pocket bulbous domain, such that there was a pairing of contractile vacuole and Golgi to be inherited by each daughter cell (Fig 3A). We were unable to identify which contractile vacuole/Golgi pair re-positioned with the new flagellum in Stages 2–4 because flagella pocket division had not yet occurred. However, there was a clear difference in volume between the old and new flagellum-associated contractile vacuole in Stage 5 (Fig 3C). Overall, this shows that as the cytoplasmic ridge ingresses to segregate the flagellar pocket bulbous domains, each bulbous domain acquires a dedicated contractile vacuole/Golgi pair, indicating that each bulbous domain has the potential to be endo/exocytically active at this point.

## Multi-copy organelles increase in number and overall volume during the cell cycle

Multi-copy organelles such as lipid bodies, glycosomes and acidocalcisomes also need to duplicate through the cell cycle. In Stage 1 cells there were 6 ± 3.6, lipid bodies, 46 ± 17.3 acidocalcisomes and 37± 8.8 glycosomes (n = 6 cells) distributed throughout the cell body (Fig 3D–3I). Between Stages 1–4 as the cell increases in size there was an increase in these multi-copy organelles (Fig 3G–3I). The number of organelles plateaued around Stage 5 and by Stage 6, there were two nascent daughter cells, and these organelles were positioned away from the division fold segregated into the regions that would become the two daughter cells (Fig 3D–3F; dashed line–division fold) and each organelle was evenly distributed (Fig 3G–3I). The total volume of each multi-copy organelle in each cell type was also calculated and between Stages 1 and 2 decreased slightly before increasing from Stage 2 to 4 (Fig 3J–3L).

## The mitotic nuclear bridge is devoid of nuclear pores during closed mitosis

Next, we examined division of the kinetoplast and nucleus (Fig 1C and 1D). The volume of the kinetoplast increased between Stages 1–4, with division into two separate kinetoplast disks completed by Stage 5 (Figs 1C and 4A). Nuclear volume and surface area increased up to Stage 5, after which the two nuclei were the same size (Fig 4B and 4C). Stage 5 cells were characterised by a mitotic nucleus with a fully intact nuclear envelope and two distinct lobes connected by a narrow nuclear envelope-bound bridge (Fig 1C and 1D; arrow) and the late mitotic nucleus was orientated perpendicular to the anterior-posterior axis of the cell (Fig 1C and 1D; Stage 5). Interestingly, in one cell with two nuclei, we observed membrane bound structures that traced a line from one nucleus to another that could represent the disassembly of the

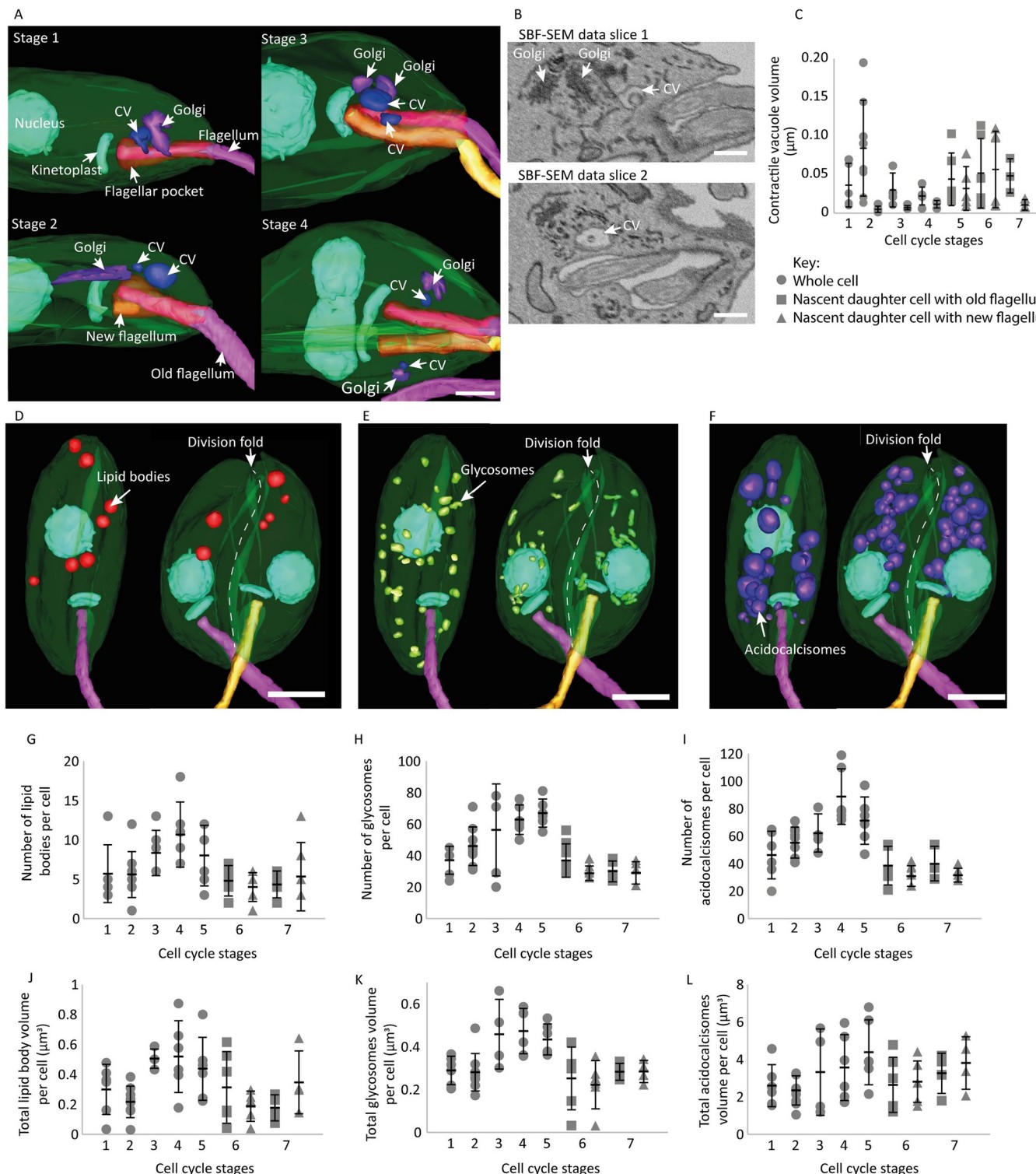

**Fig 3. Increase in multi-copy organelle volumes are seen in early cell cycle stages.** (A) 3D reconstructed Stage 1–4 cells highlighting the duplication timing and movement of the Golgi (purple) and contractile vacuole (dark blue) as the flagellar pocket duplicates (opaque red). Scale bar = 1 μm. (B) SBF-SEM data slices of the appearance of an additional contractile vacuole (CV) and Golgi next to a CV and Golgi in a Stage 3 cell. Scale bar = 500 nm. (C) Dot plot graph showing contractile vacuole volumes across the seven cell cycle stages defined. Error bars show ±standard deviation; n = 41. (D-F) 3D reconstructed Stage 1 and Stage 6 cell showing the positioning of the lipid bodies (red), glycosomes (green) and acidocalcisomes (purple) within the cell body. White dotted line along the Stage 6 cell indicates where the division fold lies and the separation of the multi-copy organelles away from this fold in the cell membrane. Scale bar = 2 μm. Dot plots

showing (G-I) the number of lipid bodies, glycosomes and acidocalcisomes per cell and (J-L) the total volume of lipid bodies, glycosomes and acidocalcisomes per cell across the seven cell cycle stages defined. n = 41. Key for 3C, 3G-L: Whole cells (circle data points), nascent daughter cell inheriting the old flagellum (square data points) and nascent daughter cell inheriting the new flagellum (triangle data points). Error bars show ±standard deviation. Total number of cells per stage: 6 cells in Stage 1, 9 cells in Stage 2, 4 cells in Stage 3, 6 cells in Stage 4, 6 cells in Stage 5, 6 cells in Stage 6 and 4 cells in Stage 7.

nuclear envelope bound bridge (Fig 4D; arrows). The kinetoplast volume and nuclear volume and surface area of Stage 2 cells was larger than that of Stage 1 cells, indicating that both kinetoplast and nucleus S-phase had likely begun in Stage 2 cells, along with new flagellum assembly and second contractile vacuole appearance.

Our data were of sufficient resolution to examine the number of nuclear pores in the nuclear envelope through the cell cycle (Fig 4E; arrows). Nuclear pores were distributed across the nuclear envelope and their number increased across the cell cycle from 27±3.5 in Stage 1 to 41±13.9 in Stage 5 cell (Fig 4F and 4G). However, given the average pore width of ~80 nm (n = 6 Stage 1 cells) and our slice thickness of 100 nm, our nuclear pore counts are likely underestimates of the total number. The average nuclear pore density (NP per nucleus surface area) was not constant throughout the cell cycle (Fig 4H). Nuclear pore density increased from 2.5±0. 6 per $\mu m^2$ in Stage 1 to 3±0.3 per $\mu m^2$ in Stage 2 (n = 15 cells) but then decreased to 1.8 ±0.62 per $\mu m^2$ during late mitosis in Stage 5 (n = 6 cells). Nuclear pore density then increased to 3.1±0.82 $\mu m^2$ in Stage 6 (n = 6 cells) at which point nuclear pore density was similar between the two daughter nuclei.

Interestingly, the nuclear bridge of the mitotic nucleus was relatively devoid of nuclear pores (Fig 4I). Given the surface area of this bridge region was 1.8 ± 2 $\mu m^2$ (n = 6 Stage 5 cells) and the average number of nuclear pores per membrane surface area was ~2±0.62 per $\mu m^2$ at Stage 5, we would expect to see ~4 pores within the bridge and in the 6 bridges examined we only saw 3 nuclear pores in total ($p$ = 0.01046, Mann-Whitney U test). This suggests there is a potential exclusion mechanism to ensure the nuclear bridge is devoid of nuclear pores or that nuclear pores are quickly disassembled at the centre of the dividing nucleus, just before this region narrows down and becomes a noticeably 'bridge-like'.

## Mitochondrion retains a complex structure throughout the cell cycle

The ER and mitochondrion are large networks that must grow and segregate during the cell cycle. The ER network was more concentrated along the periphery of the cell body and extended along its length, with clear reticulated regions connecting to the nuclear envelope (Fig 4J and 4K). In addition, we observed regions of the ER appearing to contact the cell membrane and/or the sub-pellicular microtubules (Fig 4L). We investigated these ER extensions in more detail using serial section tomography (Fig 4M and 4N). In these tomograms the ER intercalated between the sub-pellicular microtubules over an extended region (n = 7 cells) (Fig 4M and 4N). This tight associated of the ER and the sub-pellicular microtubules may play a role in ensuring ER inheritance during cell division. A similar association between the ER and the microtubules of the microtubule quartet is seen in *T. brucei* and this association is also likely important for segregating the ER during cell division [30].

The single mitochondrion forms a network of interconnected loops and branches (Fig 4O), with the volume and surface area of the mitochondrion increasing from Stage 1 to 5, peaking at Stage 5 after replication and segregation of the kDNA was complete (Fig 4P and 4Q). Mitochondrial segregation was accompanied by a reorganisation of the network in Stage 5 cells. An enlarged region of the mitochondrion was positioned along the anterior-posterior division fold, with projections of the mitochondrial network extending into the two regions of the cell

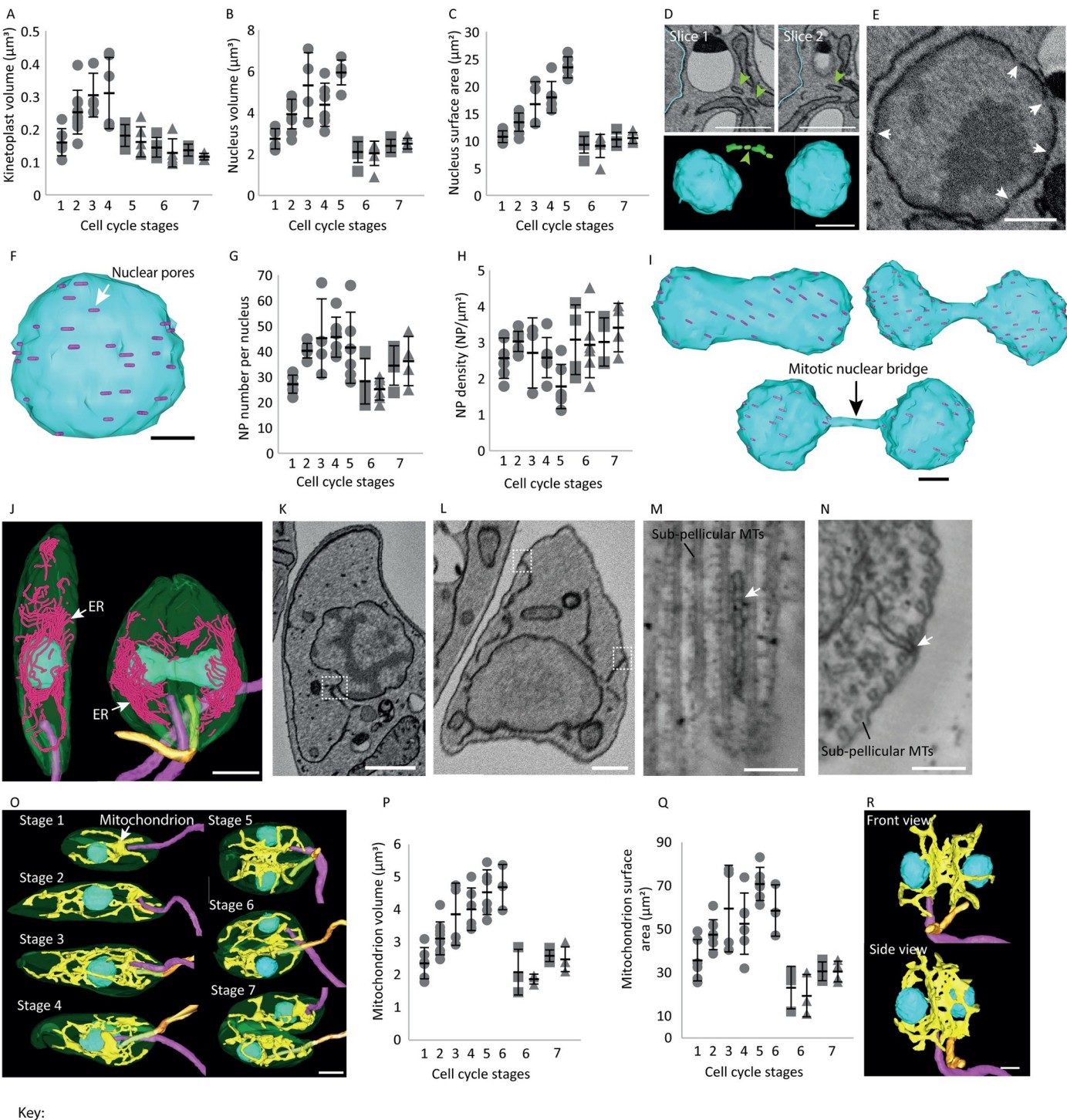

**Fig 4. Nuclear pores are excluded from the nuclear bridge during mitosis.** Dot plots showing (A) kinetoplast disk membrane volume, n = 56, (B) nucleus volume and (C) nucleus surface area per cell across the seven cell cycle stages defined. n = 52. (D) SBF-SEM dataset slices and 3D reconstruction of two nuclei (blue) and membrane bound structures (green—arrow). Scale bar = 1 μm. (E) SBF-SEM data slice of a Stage 1 nucleus showing the positioning of nuclear pores (arrows) within the nuclear envelope. Scale bar = 500 nm. (F) 3D reconstructed Stage 1 nucleus showing the distribution of nuclear pores (pink—white arrow) throughout the nuclear envelope. Scale bar = 500 nm. (G) Dot plot showing the number of nuclear pores (NP) per nucleus. n = 50. (H) Dot plot showing the nuclear pore (NP) density per nucleus. n = 50. (I) 3D reconstructed nuclei at different stages of mitosis illustrating the absence of nuclear pores (pink) along the mitotic nuclear bridge (arrow) as the nucleus progresses through mitosis. Scale bar = 500 nm. (J) 3D reconstruction of the ER (pink—white arrow) positioning in a Stage 1 and 5 cell. Scale bar = 2 μm. (K)

SBF-SEM data slice showing the ER connected to the nuclear envelope (dashed box) in a Stage 1 cell. Scale bar = 1 μm. (L) SBF-SEM data slice showing the ER close to the sub-pellicular microtubules (dashed boxes). Scale bar = 1 μm. (M-N) Serial section tomograms of the ER (arrow) intercalating between the sub-pellicular microtubules (MTs). Scale bar = 50 nm. (O) 3D reconstructed Stage 1–7 cells showing the complex mitochondrion network throughout the cell cycle (yellow). Scale bar = 1 μm. Dot plots showing (P) the mitochondrion volume and (Q) mitochondrion surface area across the seven cell cycle stages defined. Stage 6 and 7 are split into 3 and 2 categories, respectively: cells with the mitochondrion not yet segregated (circle data points), nascent daughter cell inheriting the old flagellum (square data points) and nascent daughter cell inheriting the new flagellum (triangle data points). n = 50. (R) 3D reconstructed Stage 5 cell showing the branching of the mitochondrion network (yellow). Scale bar = 1 μm. Key for 4A-C, 4F, 4G, 4K: Whole cells (circle data points), nascent daughter cell inheriting the old flagellum (square data points) and nascent daughter cell inheriting the new flagellum (triangle data points). Error bars show ±standard deviation. Total number of cells per stage: 6 cells in Stage 1, 9 cells in Stage 2, 4 cells in Stage 3, 6 cells in Stage 4, 6 cells in Stage 5, 6 cells in Stage 6 and 4 cells in Stage 7.

body that will form the daughter cells (Fig 4R). In our dataset, the mitochondrion was the last organelle to divide with only 50% (n = 6 cells) of Stage 6 cells containing two independent mitochondria. Similar late division of the mitochondrion was also seen in *T. brucei* [18,31]. Post-segregation in Stage 6 cells, the mitochondrion volume within the two nascent daughter cells was similar and increased in Stage 7 to a similar volume seen in Stage 1 cells.

### Formation of the division fold begins prior to mitotic nuclear bridge formation and is retained post cytokinesis

The shape of the cell body is defined by the subpellicular microtubule array and the plasticity of this array enables the cell body to undergo morphological changes during the cell cycle [3,4]. The first step cytokinesis is the in-folding of the cell membrane to form a division fold, as previously seen in *T. brucei* [32]. This was seen in Stage 4 and later cells with the in-folding of the cell surface occurring centrally along the anterior to posterior axis of the cell (Fig 5A). Fold formation occurred symmetrically via an in-folding from both sides of the cell body (Fig 5A and 5B; arrows). Furrow formation is first visible in Stage 6, and is more pronounced in Stage 7 cells, progressing unidirectionally from between the two flagellar pockets at the anterior end of the cell towards the posterior. Furrow progression is accompanied by the twisting of the two daughter cells away from each other, generating the characteristic posterior cell tip to tip of late cytokinetic cells (Fig 5C and 5D).

We noted that Stage 1 cells also appeared to contain a fold, which we speculated might be from a previous cytokinesis event. Stage 1 cells do not have a circular cross section and instead have a characteristic set of indentations with a central ridge along one side of the cell body that is generated by the resolution of the cytokinesis fold and furrow (Fig 5D and 5E; arrow). This fold was not present in Stage 2 cells, suggesting that as the cell body elongates microtubule rearrangement occurs to generate a cell with a circular cross-section and resolve the cytokinesis fold. As SBF-SEM does not have the sufficient resolution to visualise and reliably track sub-pellicular microtubules, we reconstructed the sub-pellicular cortex of whole cells by serial section electron tomography to reconstruct whole cells at ~2 nm resolution. Reconstructed whole cells required a large number of sequential dual-axis tomograms (S2 Fig). Thus, we focused on examining the transition from Stage 1 to Stage 2, to investigate the microtubule rearrangements that lead to ridge disappearance. We reconstructed two Stage 1 cells that did not contain a new flagellum and one Stage 2 cell where there was a short new flagellum within the flagellar pocket (Figs 5 and S2). Tracing of the microtubules from the serial section tomograms revealed a parallel array of evenly spaced microtubules, with no discontinuities such as microtubule crossovers or double layers of microtubules (Fig 5F). The microtubules lay in a weakly right-handed helical configuration, with a pitch of ~20 μm (Fig 5F). Consistent with our data and others, the cell body length increased from Stage 1 to Stage 2. Intriguingly microtubule number decreased as the cell elongated, but the overall length of the microtubules increased (Fig 5G and 5H).

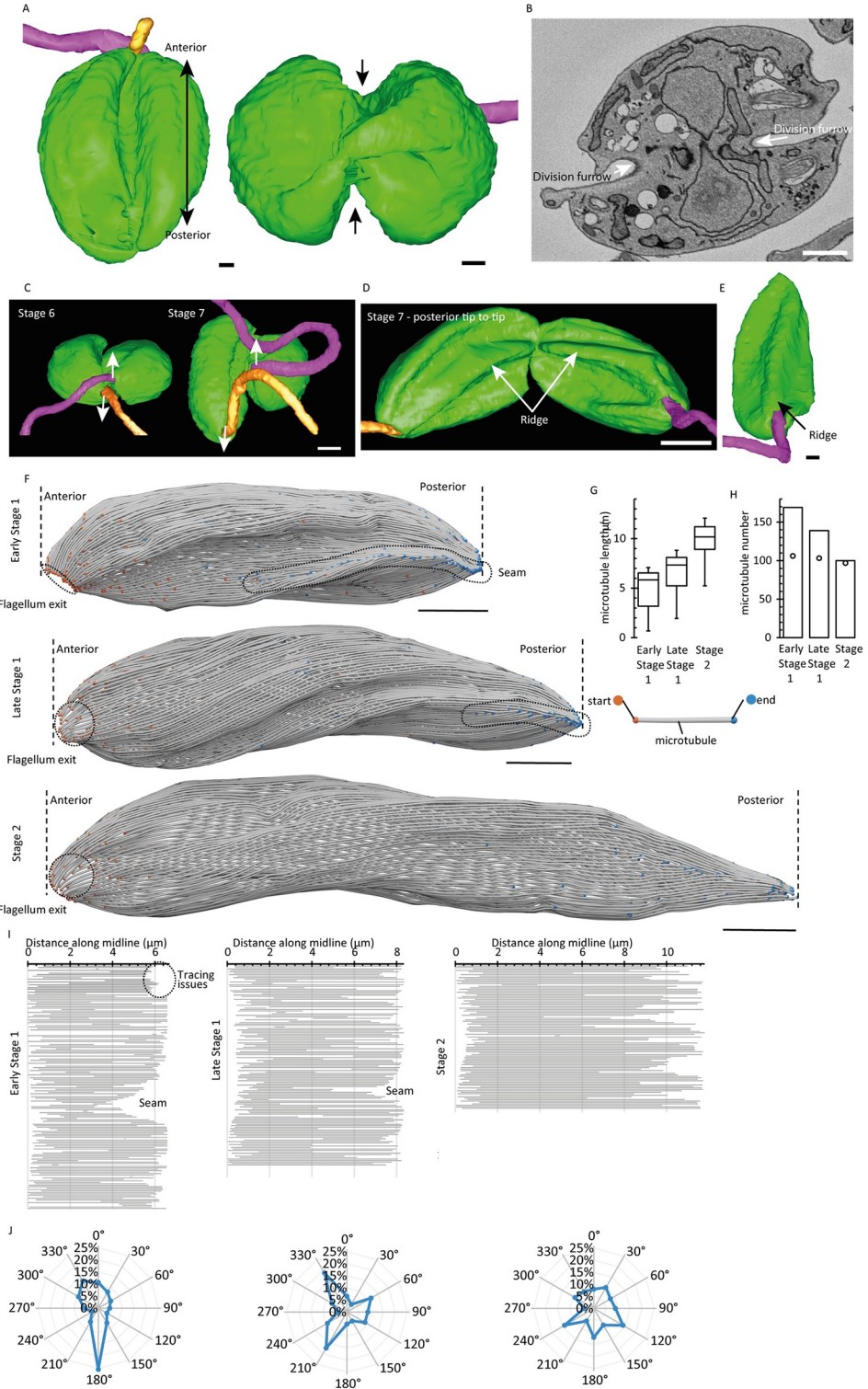

**Fig 5. Sub-pellicular microtubule organisation from whole cell electron tomography reveals a cytokinesis-derived seam.** (A) 3D reconstructed Stage 5 cell showing the division fold occurring bi-directionally and along the anterior-posterior axis of the cell body. Scale bar = 1 μm. (B) SBF-SEM data slice of a Stage 5 cell showing the in-pushing of the cell body membrane to form the division furrow (arrows). Scale bar = 1 μm. (C) 3D reconstructed Stage 6 and 7 cell showing the twisting (arrows) of the old flagellum (pink) and the new flagellum (orange) away from each other as the furrow ingresses towards the posterior end of the cell. Scale bar = 1 μm. (D) 3D reconstructed Stage 7 cell showing the

double indentation of the cell membrane to form the ridge along the anterior-posterior axis. Scale bar = 1 μm. (E) 3D reconstructed Stage 1 cell showing the presence of the ridge post abscission. Scale bar = 1 μm. (F) 3D reconstructed sub-pellicular microtubules of Stage 1 and 2 cells, from serial tomograms. In the two Stage 1 cells, a unique 'seam' of converging microtubules is visible (dashed circle). Scale bar = 1 μm. (G) Box whisker plot of sub-pellicular microtubule length distribution in each cell (Stage, S) analysed in 5F. Boxes represent the 25th, 50th and 75th percentiles and whiskers represent the 5th and 95th percentile. (H) Bar graph of number of sub-pellicular microtubules identified in each cell (Stage, S) analysed in 5F. This does not include the flagellar axoneme, microtubule quartet, or pocket and lysosome associated microtubules. Data points represent estimated minimum possible number of microtubules, from maximum cell radius. (I) Quantitative schematic of the microtubule organisation in each cell analysed in 5F, produced by 'unwrapping' the sub-pellicular microtubules around a line running along the midline of the long axis of the cell from anterior to posterior. In each plot, each line represents a microtubule and where it starts and ends along the cell midline, plotted, from top to bottom, clockwise around the cell looking posterior to anterior. The unique seams in the two Stage 1 cells are visible as neighbouring microtubules with ends far from the cell posterior. A problematic region for microtubule tracing near the Stage 1 cell posterior is indicted. (J) Plot of radial position, in 30 degree bins, where microtubules end in the Stage 1 and Stage 2 cells.

The whole cell microtubule models clearly showed the characteristic ridge and double indentation matching that formed by resolution of the division fold and furrow seen by SBF-SEM, indicating that the underlying microtubules were part of its formation. Within the ridge portion of the double indentation, we identified a line of microtubule ends (Fig 5F). To better visualise the microtubule organisation, we 'unwrapped' the microtubules around the midline of the cell along its long axis, producing quantitative schematics of where microtubules start and end within the reconstructed cell (Fig 5I). A cluster of anomalously short microtubules starting near the cell anterior but not reaching far to the cell posterior was evident (Fig 5I and 5J; seam). This was visible as a converging line of microtubule ends in the 3D microtubule organisation of the Stage 1 cells (Fig 5F and 5J). As the only discontinuity in an overall otherwise homogenous array we referred to this as a 'seam' in Stage 1 cells, which was resolved by Stage 2. Importantly this seam is at the correct position to represent a 'scar' of the line of furrow ingression from the previous cell cycle.

## Cell body volume occupied by lipid bodies was 6-fold higher in sand fly derived promastigotes

We wanted to extend our analysis to *Leishmania* promastigotes in the sand fly, as they experience the natural environment with different energetic and metabolic pressures to those in culture, which could be reflected in their 3D cellular organisation. We chose to examine parasites from a late stage infection at the stomodeal valve, as this gives a good number of parasites which are likely furthest removed from *in vitro* log phase promastigotes, giving the most extreme of comparisons. We analysed a 10-day post infection *L. mexicana* infected sand fly midgut SBF-SEM dataset containing 218 slices, which covered part of the stomodeal valve [33] (Fig 6A). Every cell examined (n = 89) was in Stage 1 based on our *in vitro* classification, as they only had one flagellum, suggesting that these cells were in stationary phase. In total, 10 random cells whose whole cell body and flagellum were laying within the dataset and their flagellum not attached to the stomodeal valve, were segmented and the volume and number of organelles—the flagellum, flagellar pocket, mitochondrion, acidocalcisomes and lipid bodies–were analysed (Fig 6B and 6C). Glycosomes, contractile vacuole, Golgi and ER were partially reconstructed in some cells but were not included in this analysis because of low contrast staining leading to lower resolution of the SBF-SEM dataset captured from a sand fly infection, compared with the *in vitro* promastigote dataset.

Previous studies have defined different promastigote stages in sand fly infections using morphological criteria including cell body and flagellum length [34,35]. Using these criteria, we identified ten leptomonad promastigotes in our dataset ranging from 5.6–13.3 μm in cell

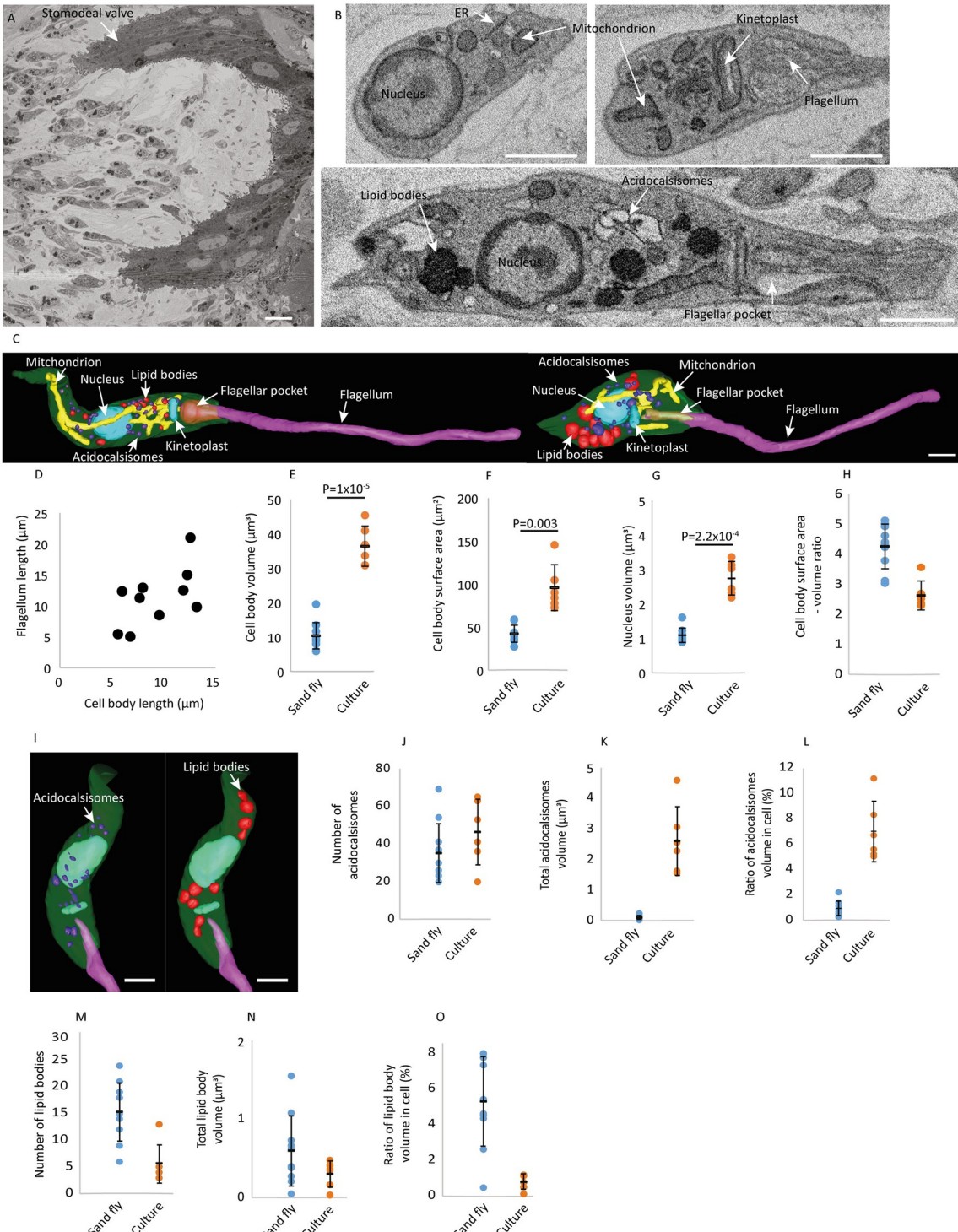

**Fig 6. Culture and sand fly derived promastigotes differ in organelle volume and spatial organisation.** (A) SBF-SEM data slice showing the cross section of the sand fly stomodeal valve. Scale bar = 5 μm. (B) SBF-SEM data slice of sand fly derived cells highlighting the key organelles identified. Scale bar = 1 μm. (C) 3D model of leptomonad promastigote cells showing the spatial organisation of organelles within the cell body. Organelles are modelled using the following colours: cell body (green), flagellum (pink), nucleus (blue), kinetoplast (blue), mitochondrion (yellow), acidocalcisomes (dark purple), lipid bodies (red) and the flagellar pocket (opaque deep red). Scale bar = 1 μm. (D) Scatter plot of cell body length against flagellum length. Each data point represents one cell. n = 16. Dot plot showing (E) cell body volume (F) cell body surface area (G) nucleus volume and (H) cell body surface area to volume ratio between the

sand fly derived cells and culture derived (Stage 1) cells. n = 16, t-Test. (I) 3D models of leptomonads showing the spatial organisation of acidocalcisomes (purple) and lipid bodies (red) throughout the cell body. Scale bar = 1 μm. Dot plot showing (J) number of acidocalcisomes per cell (K) total acidocalcisomes volume per cell and (L) ratio of acidocalcisomes volume in cell between the sand fly derived cells and culture derived (Stage 1) cells. n = 16. Dot plot showing (M) number of lipid bodies per cell (N) total lipid body volume per cell and (O) ratio of lipid body volume in cell between the sand fly derived cells and culture derived (Stage 1) cells. n = 16. Error bars show ±standard deviation. 6 Stage 1 culture derived cells and 10 sand fly derived cells were analysed.

body length, with flagella ranging from 5–21 μm in length (Fig 6C and 6D). The mitochondrion structure remained complex in the leptomonad, resembling that seen in the *in vitro* promastigotes (Figs 6C and 4M). Overall, the promastigotes found in the stomodeal valve of the sand fly were significantly smaller in cell body volume, surface area and nucleus volume, while cell body surface to volume ratio was significantly higher, in comparison to Stage 1 *in vitro* promastigotes (Fig 6E–6H). Acidocalcisomes and lipid bodies were also distributed throughout the cell body in the leptomonad promastigotes, similar to the culture promastigotes (s 6I, 3D-F). When we examined the numbers and volumes of acidocalcisomes and lipid bodies we also saw differences between the leptomonad and *in vitro* promastigotes. The number and total volume of acidocalcisomes was lower in the sand fly promastigote cells, representing 0.9% of the total cell body volume in comparison to 7% for the *in vitro* cells (Fig 6J–6L). In contrast to acidocalcisomes, lipid bodies increased in number and in total volume, representing 5% of the total volume of the cell body in comparison to the *in vitro* cells where lipid bodies only represented 0.8% (Fig 6M–6O). This suggests differences in metabolism between the *in vitro* and sand fly parasites, with an increase in lipid storage in the sand fly parasites that correlates with transcriptomic data [36].

## Discussion

Our study has provided an in-depth quantitative and qualitative analysis of the *L. mexicana* promastigote cell cycle, defining a morphological framework of seven distinct cell cycle stages in which ultrastructural rearrangements occur in a defined sequence for accurate cell division (summarised in Fig 7). We showed that initiation of assembly of the new flagellum is one of the first events in the *Leishmania* cell cycle, while at the end of the cell cycle the resolution of the sub-pellicular microtubules leaves a characteristic seam defined by the ends of shorter microtubules.

In addition to ultrastructural rearrangements, there is an orchestrated set of movements to ensure the segregation of organelles between the two daughter cells, with for example the changes to the mitochondrion organisation during the cell cycle. For those organelles connected to and associated with the basal body, including the flagellum, flagellar pocket, kinetoplast and mitochondrion, the positioning of the cytokinesis furrow between the basal bodies ensures the segregation of these organelles between the two daughter cells [8,22,37]. Moreover, the ER connects to the nuclear envelope and was shown to intercalate between the sub-pellicular microtubules; the combination of the positioning of the two nuclei post-mitosis and the anchoring of the ER to the microtubules would ensure its segregation. The segregation of both the ER and flagellar pocket between the two daughter cells will likely ensure the inheritance of the Golgi. Multi-copy organelles such as glycosomes were shown to increase in number before cell division and they are likely present in sufficient numbers that their random segregation would ensure each daughter inherited a sufficient number, while new copies could be generated by the ER. While, we cannot exclude a segregation mechanism for these multi-copy organelles through an association with a specific cytoskeletal element, we did not observe any such connection.

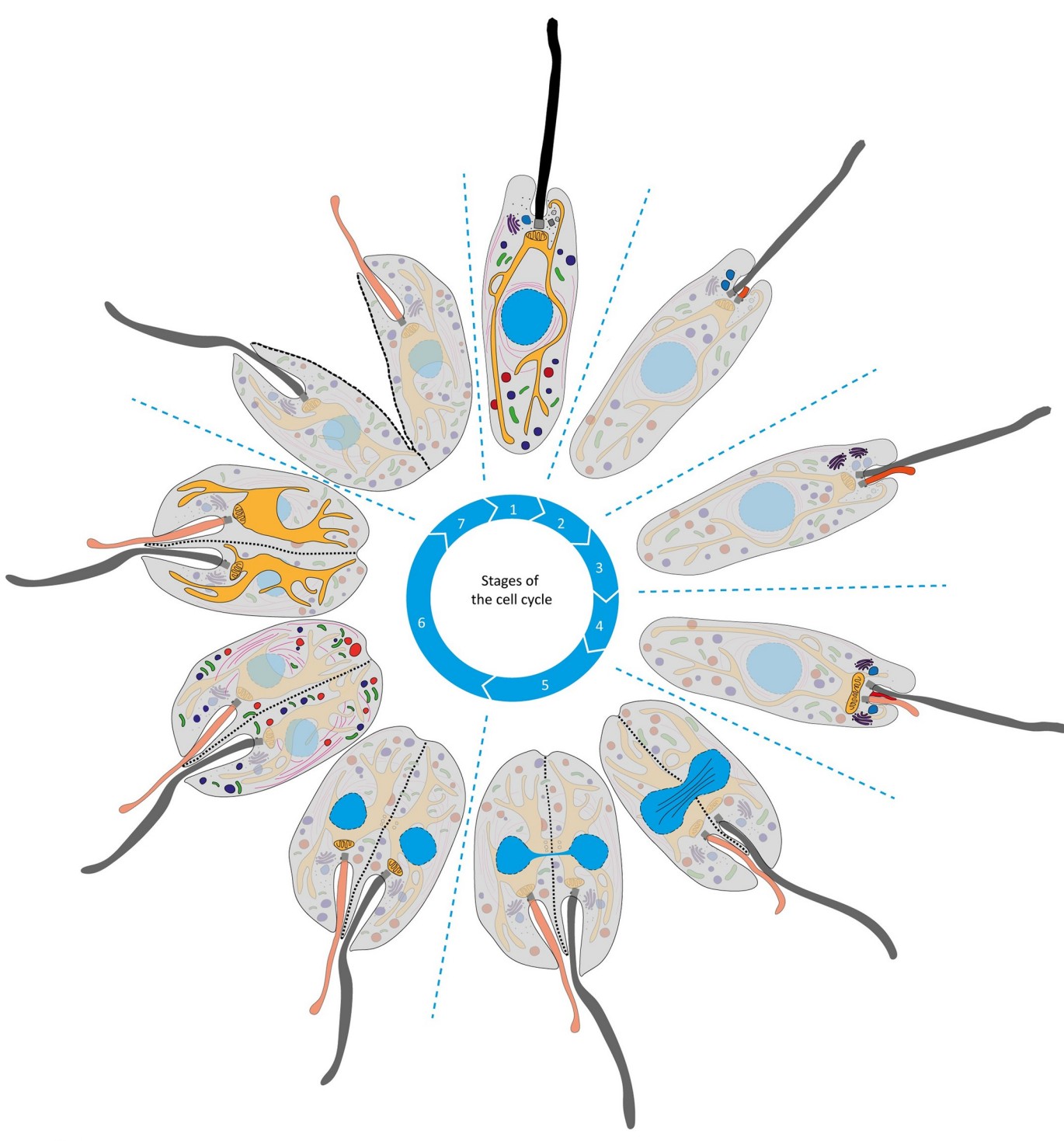

**Fig 7. Illustration of the morphological events occurring during the cell cycle of *L. mexicana* promastigote cells as defined in Fig 1.** Fig 7 does not represent the approximate timing of the cell cycle events from Fig 1E. Organelles are modelled using the following colours: cell body (grey), old flagellum (black), new flagellum (orange), nucleus (blue), kinetoplast (blue) mitochondrion (yellow), acidocalcisomes (dark purple), lipid bodies (red), glycosomes (pale green), contractile vacuole (dark blue), Golgi (purple), ER (pink), cytoplasmic spur in the flagellar pocket (red). Black dashed line represents the division fold and furrow forming along the anterior-posterior axis of the cell body. Blue dashed lines on the nuclear envelope represent nuclear pores. Stage 1 represents a G1 cell. The cell enters S-phase in Stage 2 and a new flagellum is assembled within the flagellar pocket and the contractile vacuole duplicates. The new flagellum grows in length and

extends out of the flagellar pocket in Stage 3. The Golgi duplicates and the nucleus enters mitosis. In Stage 4 the kinetoplast begins to undergo division. During Stage 5, the kinetoplast has divided and the nucleus is in the process of nuclear division. A cytoplasmic spur forms within the existing flagellar pocket to divide the flagellar pocket between the old and new flagellum. One contractile vacuole and Golgi appears either side of the dividing flagellar pocket. A division fold forms along the anterior-posteriors axis, pinching the cell body into two nascent daughter cell morphologies. In Stage 6, two separate nuclei and kinetoplast are present and the flagellar pocket has divided. Lipid bodies, acidocalcisomes, glycosomes and the endoplasmic reticulum segregate between the two nascent daughter cells. The mitochondrion is the last organelle is segregate. During Stage 7, the cell body undergoes cytokinesis which gives rise to one daughter cell which inherits the old flagellum and one daughter cell inheriting the new flagellum.

In the cell cycle, the first ultrastructural change we identified was the assembly of the new flagellum within the flagellar pocket and the appearance of a second contractile vacuole, which also coincided with an increase in nuclear and kinetoplast volume, indicating that S phase had likely begun. Based on our ergodic analysis this occurred at 0.3 u through the cell cycle, which is comparable with our previous light microscopy studies on the *L. mexicana* cell cycle which showed that nuclear and kinetoplast S phase begins at 0.38 u [4]. From this and being aware of the limitations of previous experimental approaches used, it is likely that S phase, new flagellum assembly, new microtubule quartet and contractile vacuole duplication are initiated at a similar time in the cell cycle. This is consistent with our previous tomographic analysis of the *Leishmania* flagellum attachment zone [8]. The reproducible progression of the cell cycle suggests there are a series of molecular triggers that control organelle duplication but these are still to be elucidated for *Leishmania*. The maturation of the basal body early in the cell cycle of both *L. mexicana* and *T. brucei* suggests that cell cycle initiation in these organisms is similar and potentially reliant on common regulatory molecules and mechanisms such as the mitotic arrest deficient 2 protein (MAD2) and the cdc2 related protein kinase 7 (CRK7), and the monitoring of the cellular metabolic status [38,39].

Previous analyses of the cell cycle of different *Leishmania* species by light microscopy have shown variation in whether kinetoplast division precedes nucleus division or vice versa [3,4,16,40,41]. In this study, we observed the kinetoplast starting to divide before the nucleus as we classed late mitotic cells as having one nucleus as they were still connected via the mitotic bridge membrane. This bridge is not easily visible via light microscopy methods, so there is the potential for such a cell to appear to have two separate nuclei when imaged using DNA stains and light microscopy [4]. Yet, both these approaches are based purely on morphology and the regulators and the timing of their appearance remain unknown in *Leishmania*.

In organisms including *Leishmania*, which divide their nucleus via closed mitosis, the nuclear envelope needs to be resolved and segregated between two independent nuclei [42,43]. We found that, in the narrow bridge formed between the mitotic nuclei there was a reduction in the number of nuclear pores. This aligns with recent work in *S. pombe* that showed the composition of the nuclear pore complex changed during nuclear envelope resolution; nuclear pore exclusion in *Leishmania* is therefore likely important for the resolution of mitosis [44]. Moreover, in *T. brucei* the protein Closed Mitosis Protein 1 was shown to be important for nuclear envelope resolution and localised to mitotic nuclear bridge on the nuclear envelope and this may be related to the nuclear pore exclusion phenomenon [45]. Together this suggests that mechanisms to create differentiated regions of nuclear envelope are common to organisms that undergo closed mitosis.

The *Leishmania* cell body undergoes several morphological changes during differentiation from a promastigote form to an amastigote form [4,5]. Little is known about how the sub-pellicular microtubule array undergoes re-modelling to accommodate these cell body changes. However, the preference for long, near whole cell length, microtubules in Stage 2 cells indicates that cell shortening for cytokinesis and differentiation to amastigotes must involve

microtubule depolymerisation. Perhaps the shorter amastigote cell body is generated by division. Unlike for *Leishmania*, we do not have whole cell 3D mapping of all sub-pellicular microtubules for any other trypanosomatid. However, light microscopy evidence suggests that, during cell division, *T. brucei* and *T. cruzi* add new microtubules to the existing subpellicular array rather than depolymerisation of microtubules [32,46–48].

We discovered that pre-abscission cells (Stage 7) have a double indentation and ridge of the cell membrane, which is present in daughter cells after the cytokinesis furrow has processed along the anterior to posterior axis. This double membrane indentation and ridge is also present in Stage 1 cells but was resolved by Stage 2. In Stage 1 cells, we observed by electron tomography a seam along which short microtubule ends meet and this discontinuity in the otherwise uniform sub-pellicular microtubule array is likely to be partially responsible for the ridge and double membrane indentation of the cell membrane we observed in Stage 1 cells. Consistent with this seam contributing to the double membrane indentation these short microtubules were resolved by Stage 2, when the ridge and indentations also disappear. Overall, this points to the preferred growth of microtubules in the cleavage furrow from anterior to posterior as the cytokinesis furrow is formed. The retention of these indentations in the cell body post cytokinesis and the microtubule seam is analogous to what has been described in budding yeast where the mother retains a bud scar and the daughter cell a birth scar post cytokinesis, though it has yet to be determined in *Leishmania* whether a permanent molecular remnant is associated with this seam [49].

Our data showed that the first step in flagellar pocket division was the invasion of the cytoplasmic ridge between the old and new flagella and this ridge was also observed in *T. brucei*, suggesting its formation is likely important for the flagellar pocket division in many kinetoplastid parasites. In *T. brucei* the new flagellum rotates around the old flagellum, bending the flagellar pocket membrane around the microtubule quartet, which contributes to flagellar pocket division [22]. Yet, we showed that this rotation does not occur in *Leishmania*; therefore, the invasion of the cytoplasmic ridge is likely the dominant mechanism for the division of the flagellar pocket bulbous domain. This difference in flagellar pocket division between *Leishmania* and *T. brucei* reflects their liberform and juxtaform morphologies and the requirement of *T. brucei* to position its new flagellum to the posterior of the old one, with the rotation a juxtaform adaptation not found in the ancestral form [7]. At the point the cytoplasmic ridge was observed in the flagellar pocket (Stage 4), the new flagellum had already extended beyond the cell body. This means the new flagellum shares the flagellar pocket neck region and exit point with the old flagellum. The duplication of the flagellar pocket and neck structures must therefore occur through a semi-conservative mechanism, with new molecular components integrated into an existing structure. Moreover, this suggests that flagellar pocket segregation likely occurs through a two-step process, with the bulbous domain dividing first through the continual invasion of the cytoplasmic ridge, followed the division of the neck region.

*Leishmania* undergo a series of morphologically and functionally distinct developmental stages in the sand fly [50]. While in their natural vector *Leishmania* spp. are exposed to nutrient starvation and stress, and must utilise a range of different carbon sources, which differ from the promastigotes growing in culture [36,51]. Our study shows that while there are changes to organelle number the overall cell morphology and architecture is retained between sand fly and culture promastigotes. The promastigotes from a late stage infection had a large number and volume ratio of lipid bodies, and this correlates with previous transcriptomic work that showed the upregulation of fatty acid biosynthesis pathways in the late stages of *Leishmania* development within the sand fly [36,52]. Moreover, as fatty acids are a major carbon source of the amastigote stage within the macrophage parasitophorous vacuole, this suggests that there is pre-adaptation for survival in the mammalian host [53].

SBF-SEM is becoming a routine method to visualise whole cells and the changes that occur during the cell cycle. In this study we have combined our previous knowledge of the *Leishmania* cell cycle to provide a detailed quantitative and qualitative analysis of organelle duplication and inheritance patterns, and the morphological changes that occur during cytokinesis to ensure two identical daughter cells within the liberform morphological supergroup. These foundational insights will be used to shape future molecular cell biology studies of *Leishmania*.

## Materials and methodology

### *Leishmania mexicana* culture

*L. mexicana* (WHO strain MNYC/BZ/1962/M379) promastigotes were grown at 28˚C in M199 medium with 10% foetal calf serum, 40 mM HEPES-HCl (pH 7.4), 26 mM $NaHCO_3$ and 5 μg/ml haemin.

### Infection of *L. mexicana* in the sand fly

Sand fly *L. mexicana* infection was carried out as described in [55]. Female *Lutzomyia longipalpis* were fed heat-inactivated sheep blood containing log phase *L. mexicana* promastigotes at a concentration of 2 x $10^6$ cells/ml, through a chick-skin membrane. Blood engorged females were separated and maintained at 26˚C and high humidity with free access to a 50% sugar solution, with a 14-h light/10-h dark photoperiod. They were dissected on day 10 after a blood meal, and the dissected guts were fixed for 24 h at 4˚C in Karnovsky fixative (2.5% glutaraldehyde and 2% paraformaldehyde in 0.1 M cacodylate buffer (pH 6.9), transferred to 0.1 M cacodylate buffer with 2.7% glucose washing solution and kept at 4˚C until further processing.

### Resin embedding for serial-block face scanning electron microscopy (SBF-SEM)

*Leishmania*-infected sandfly midguts fixed as described above were washed and kept in 0.1 M cacodylate buffer (pH 6.9) at 4˚C until post-fixation. *L. mexicana* promastigotes (C9T7) were grown to exponential phase (1 x $10^7$ cells/ml) as described above and then fixed in growth medium by the addition of glutaraldehyde for a final concentration of 2.5%. Samples were then spun and fixed in 2.5% glutaraldehyde in 0.1M phosphate buffer and washed in the same buffer immediately before post-fixation. After primary fixation with aldehydes, all samples were post-fixed in 1% osmium tetroxide and 1.5% potassium ferricyanide in 0.1 M buffer (phosphate or cacodylate, for in vitro promastigotes and midguts, respectively), for 1 hour in the dark. Samples were then washed in $ddH_2O$, incubated in freshly prepared 1% thiocarbohydrazide for 20 min in the dark, washed in $ddH_2O$, and incubated in 2% osmium tetroxide in $ddH_2O$ for 30 min in the dark. Samples were washed again in $ddH_2O$ and incubated in 1% uranyl acetate in $ddH_2O$ overnight, at 4˚C and in the dark. Samples were washed with $ddH_2O$, dehydrated in an ethanol series and embedded in TAAB 812 Hard resin (TAAB).

### Serial-block face scanning electron microscopy

The tips of resin blocks containing samples were trimmed and mounted onto aluminium pins using conductive epoxy glue and silver dag, and then sputter coated with a layer (10–13 nm) of gold in an Agar Auto Sputter Coater (Agar). Before SBF-SEM imaging, ultrathin sections (70 nm) of the block face were examined in a Jeol JEM 1400 Flash transmission electron microscope (JEOL), to verify sample quality. Samples were then imaged in a Merlin VP compact high resolution scanning electron microscope (Zeiss) equipped with a 3View 2XP stage (Gatan-Ametek), an OnPoint back-scattered electron detector (Gatan-Ametek) and a focal

charge compensation device (Zeiss). The following imaging conditions were used: 1.8kV, 20 μm aperture, 100% focal charge compensation, 5nm pixel size, 2–4 μs pixel time, 100nm (in vitro promastigotes) or 75nm (midguts) section thickness.

## Serial-block face scanning electron microscopy segmentation and data analysis

Data were processed using the IMOD software package [56]. Briefly, image stacks were assembled, corrected (for z scaling and orientation) and aligned using eTOMO, and 3D models were produced using 3dmod. Whole individual promastigote cells in varying stages of the cell cycle were identified and trimmed from the dataset and converted into single.mrc files. Within our sand fly stomodeal valve SBF-SEM dataset, we were biased in our selection process as we only identified and analysed promastigote cells whose cell body and flagellum were entirely within the dataset. Incomplete sand fly derived promastigote cells and cells attached to the stomodeal valve (haptomonad promastigotes) were discarded from our analysis. Metacyclic promastigotes are defined based on the morphological criteria where the flagellum length is at least twice the length of the cell body (cell body length < 14 μm)[57]. Our sand fly stomodeal valve SBF-SEM dataset comprised of 218 slices at 75 nm thickness per slice therefore we were bias against finding complete metacyclic promastigotes whose cell body and flagellum were entirely in our dataset. Organelles were manually segmented as individual objects and identified based on specific ultrastructural characteristics (S1 Fig) and modelled following the outer membrane of each organelle. Source data can be found in S1 Table. The volume and surface area of organelles were automatically generated from the object's surface rendering. The kinetoplast volume was a measurement of the kinetoplast membrane surrounding the kDNA. Segmentation of the Golgi was challenging due to dense packing and stain density of the Golgi stacks, therefore segmentation of the Golgi stacks were combined rather than individual stacks. The volume of the nucleus included the nucleolus. Nuclear pores were identified as gaps in the nuclear envelope. The mitotic nuclear bridge was characterised by the pinching of the nuclear cytoplasm to form two distinct nuclear lobes on either side of the mitotic nuclear bridge. The length of the cell body was measured using the Slicer Window to adjust the X, Y and Z axis of the ZaP Window so the whole of the posterior-anterior axis of the cell body could be viewed in one window and cell body length measured. The SBF-SEM data set shows the entire ultrastructure of the new flagellum inside the flagellar pocket therefore the length of the new flagellum was measured entirely rather than only the external portion of the new flagellum. The statistical analysis tests (Mann-Whitney test and t-Test) were calculated in Microsoft Excel.

## Electron tomography sample preparation

*L. mexicana* promastigote cells were fixed in culture with 2.5% (v/v) glutaraldehyde (TAAB, Aldermaston, UK) for 5 min. Cells were then washed and resuspended in 200 mM phosphate buffer (pH 7.0) with 2.5% glutaraldehyde and 2.0% paraformaldehyde for 2 h. The pellet was washed, post-fixed with 1% osmium tetroxide (Amsbio, Abingdon, UK) for 2 h and stained *en bloc* with 2% uranyl acetate (Amsbio, Abingdon, UK) for 2 h, then dehydrated in an ethanol series and embedded in Agar 100 resin (Agar Scientific, Stansted, UK). Sections with a nominal thickness of 300 nm were cut using an ultramicrotome and a 35 DiATOME diamond knife and collected *via* a water bath onto slot electron microscopy grids with a Formvar membrane and stained with Reynolds lead citrate (TAAB, Aldermaston, UK) for 2 min. 48 serial sections were collected across two grids, from which a low magnification approximate map was made to identify whole cells lying near-perpendicular to the plane of sectioning and lying wholly within the sectioned volume (S2 Fig).

## Electron tomography data capture

Transmission electron micrographs of tilt series for tomographic reconstruction were captured on a Tecnai TF-30 at 300 kV electron acceleration voltage using a 2048×2048 Gatan CCD, at a magnification of 23.2 Å/px. Images were captured at 1˚ steps from −65˚ to +65˚ using Serial EM, recording the location of each field of view using the navigator (coordinate recording) function and location map images. Where necessary, up to three overlapping fields of view were captured per section to cover the entire cell. The sample was then removed, the grid rotated 90˚ in the sample holder, the navigator coordinates remapped to account for sample rotation, the same fields of view identified assisted by the navigator function, then precise field locations identified by registration of a test image to 90˚ rotated copy of the map images. A second tilt series was then captured, with the tilt axis perpendicular to the first–with the exception of a few sections where electron beam-induced damage led to loss of that grid region prior to capture of the second axis.

## Electron tomography reconstruction

Tomographic reconstruction of individual tilt series were made using the "Build Tomogram" project function of eTomo, part of IMOD. Fiducialless reconstructions were made of each tilt axis, before aligning the tomogram from the second tilt axis to the first, and averaging into a dual axis tomogram. The second axis typically showed an electron beam-induced sample shrinkage of ~10%. For sections captured with multiple overlapping tomographic volumes, these were subsequently montaged using the "Blend Montages" project function of eTomo, using the field of view captured first as the reference volume for alignment. Finally, the sample sections within the tomographic volumes were flattened using the "Flatten Volume" tool of eTomo.

For alignment of serial tomograms, extremely precise local alignment was necessary to track sub-pellicular microtubules separated by only ~20 nm. The "Join Serial Tomograms" project function of eTomo was found to be inadequate: Subtle section warping, likely from a combination of variation in electron beam induced sample shrinkage and local mechanical deformation from ultramicrotomy depending on local sample properties, necessitated custom alignment. Matched reference points at the top and bottom of sequential tomographic volumes, typically microtubules in flagella, sub-pellicular microtubules in neighbouring cells and mitochondrial cristae, were marked across the tomogram. Delaunay triangulation was used to split the image into triangular regions, and the position of the bounding reference points of each triangular region in the two tomograms were used to calculate the necessary affine transformation to warp that region of one tomogram to align with the other. This warping was applied to each triangle in the plane parallel to the section at all depths (i.e. in x, y, without affecting z), using custom ImageJ macros.

To align all sections, a central reference tomographic volume was selected. In the tomograms above and below that section, reference points were identified, triangulated and the neighbouring tomogram warped to align it to the reference section. Working up and down, reference points to the next section were identified, triangulated and the neighbouring tomogram warped, etc. For each warping step, reference points from structures in neighbouring cells were strongly prioritised over structures in the cell of interest, to avoid biasing subsequent joining of traced sub-pellicular microtubules between sections. Finally, the tomographic volumes from sequential sections were stacked into a single volume for the whole cell, with an estimated 10 to 20 nm depth lost between sections from the action of the diamond knife.

## Sub-pellicular microtubule modelling from tomographic volumes

Microtubules in the sub-pellicular array were traced and a 3D model generated using 3Dmod, part of IMOD. First, sub-pellicular microtubules were traced as open contours in the tomographic volumes from each serial section separately. Comprehensive identification of microtubules was confirmed by the expected regular ~20 nm spacing across the cell. Microtubule tracing was possible in all regions except a small volume of the anterior of cell 1, where a region of the sub-pellicular array lay perpendicular to the tilt axis and perpendicular to the imaging plane of the camera leading to insufficient z resolution to identify individual microtubules.

Traced microtubules were joined across serial sections based on both proximity, i.e. the microtubule is in the same place in the two sections, and trajectory, i.e. making that join does not introduce a sharp kink or bend in the microtubule. A consistent approach was used to avoid bias. First, the traces of microtubules which could be unambiguously identified in both sections were joined. In order of preference: 1) based on distinctive nearby non-sub-pellicular microtubule features, like nearby ER or mitochondrial structures. 2) based on distinctive folds or curves in the sub-pellicular array, particularly where there is a sharp bend with two or three protruding microtubules. 3) based on variation in microtubule spacing, where a distinctive narrow or wide spacing can be seen in both sections. Having made joins between sections at these unambiguous sites, this provides 'landmark' joins at many positions around cell perimeter. The intervening microtubules can then be joined. On the basis that microtubule starts and ends were rare within the tomographic volumes, joins were made which minimised the number of necessary microtubule starts and ends falling in the lost volume between sections.

To map microtubule lengths cell-wide, microtubules in each serial section tomogram needed to be correctly joined across the 7 to 11 300 nm serial sections, between which a small depth (~10–20 nm) is lost, ~5% of the cell volume. We therefore carefully considered our strategy for microtubule joining. Based on the cell circumference and even ~20 nm microtubule spacing, we expected ~100 to 150 microtubules around the widest point of each cell. As ~5% of the cell volume is lost between sections, microtubule starts and ends lost in this lost volume should be 20× rarer than starts and ends visible within sections, yet in-section starts and ends were rare. Therefore, microtubule starts and ends lost between sections must be extremely rare, implying the subpellicular microtubules must be long and the correct between-section joins should introduce a minimum number of microtubule starts and ends. Following this strategy, and using advanced section-section alignment, we build high confidence model of all microtubules in the three cells.

Therefore, where the same number of microtubules lay between these landmarks in both sections, this implies no microtubule starts/ends in the between section space and therefore join the traces of these microtubules, while monitoring for anomalously large disjoints. Where the number of microtubules between two landmarks differed between two sections, then a best effort from proximity and trajectory was made to identify which microtubule(s) started/ended within the section. Only very close to the cell anterior or posterior were cases identified where multiple nearby microtubules seemed to start/end within the between section space–these were more likely to be erroneously joined, however any error here does not affect the major conclusions.

## Negative staining protocol

For preparation of whole-mount promastigote cytoskeleton samples, 8 ml of log phase promastigotes ($1 \times 10^7$ cells/ml) were harvested by centrifugation (800g for 10 min) and resuspended in 500 μl of M199. 8 μl of cell resuspension was added to a glow-discharged formvar and carbon-coated 200 mesh nickel grid and left for 2 min. Promastigotes attached on the

formvar membrane were treated with 1% IGEPAL in PEME (0.1 M PIPES, pH 6.9, 2 mM EGTA, 1 mM MgSO$_4$, 0.1 mM EGTA) for 5 min, fixed with 2.5% glutaraldehyde in PEME for 10 min and stained with 1% aurothioglucose in ddH$_2$O. For preparation of cell body microtubule depolymerised cytoskeleton samples, samples were treated with 300 mM CaCl$_2$ in PEME for 2 min and washed twice with PEME after the IGEPAL treatment. The samples were observed using a Jeol JEM-1400 Flash transmission electron microscope operating at 120 kV and equipped with a OneView 16-megapixel camera (Gatan/Ametek, Pleasanton, CA).

## Supporting information

**S1 Movie. Serial block face scanning electron microscopy dataset of *cultured L. mexicana* promastigote cells.** Scale bar = 5 μm.
(MP4)

**S2 Movie. 3D reconstructed Stage 1 *L. mexicana* promastigote cell.** Organelles are modelled using the following colours: cell body (green), flagellum (pink), nucleus (blue), kinetoplast (blue) mitochondrion (yellow), acidocalcisomes (dark purple), lipid bodies (red), glycosomes (pale green), contractile vacuole (dark blue), Golgi (purple), ER (blue), flagellar pocket (deep red). Scale bar = 2 μm.
(MP4)

**S3 Movie. 3D reconstructed Stage 6 *L. mexicana* promastigote cell.** Organelles are modelled using the following colours: cell body (green), old flagellum (pink), new flagellum (orange), nucleus (blue), kinetoplast (blue) mitochondrion (yellow), acidocalcisomes (dark purple), lipid bodies (red), glycosomes (pale green), contractile vacuole (dark blue), Golgi (purple), ER (pale blue), flagellar pocket (deep red). Scale bar = 2 μm.
(AVI)

**S1 Fig. SBF-SEM key to image segmentation used to define organelle structure.** Each organelle was 3D reconstructed in a separate colour as defined in the key.
(TIF)

**S2 Fig. Summary of the fields of view and reconstruction strategy for whole-cell tomography.** Fields of view on the serial sections for (A) the early G1 (16 tomograms, 11 serial sections), (B) the mid-G1 (18 tomograms, 8 serial sections) and (C) the S phase cells (17 tomograms, 7 serial sections). In each, sections are named with c indicating cell number, s indicating section number and l indicating the range of field of view location numbers–with each location being a tomogram. The arrows indicate the approximate tilt axes, with one arrow for sections captured with a single tilt axis, and the section name in bold is the centre section to which the other sections were aligned. A small region in c01s10l01 in which microtubule tracing was not possible and thus ends not mapped is circled, due to microtubule orientation in the single axis tomogram. (D) Summary of the strategy for accurate reconstruction of whole cells by tomography.
(TIF)

**S1 Table. Complete *L. mexicana* culture and sand fly derived promastigote cell data.**
(XLSX)

## Acknowledgments

We would like to thank The Oxford Brookes Centre for Bioimaging for assistance in carrying out this project.

## Author Contributions

**Conceptualization:** Richard John Wheeler, Sue Vaughan, Jack Daniel Sunter.

**Formal analysis:** Molly Hair, Richard John Wheeler, Sue Vaughan, Jack Daniel Sunter.

**Funding acquisition:** Ryuji Yanase, Richard John Wheeler, Petr Volf, Sue Vaughan, Jack Daniel Sunter.

**Investigation:** Richard John Wheeler, Jovana Sádlová.

**Methodology:** Ryuji Yanase, Flávia Moreira-Leite, Richard John Wheeler, Jack Daniel Sunter.

**Project administration:** Richard John Wheeler, Petr Volf, Sue Vaughan, Jack Daniel Sunter.

**Resources:** Jovana Sádlová, Petr Volf.

**Supervision:** Richard John Wheeler, Sue Vaughan, Jack Daniel Sunter.

**Visualization:** Molly Hair, Flávia Moreira-Leite, Richard John Wheeler, Sue Vaughan, Jack Daniel Sunter.

**Writing – original draft:** Molly Hair, Flávia Moreira-Leite, Richard John Wheeler, Sue Vaughan, Jack Daniel Sunter.

**Writing – review & editing:** Molly Hair, Ryuji Yanase, Flávia Moreira-Leite, Richard John Wheeler, Jovana Sádlová, Petr Volf, Sue Vaughan, Jack Daniel Sunter.

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
