## [Decision Letter · Decision Letter 0]

17 Jan 2024

Dear Dr. Sunter,

Thank you very much for submitting your manuscript entitled “Whole cell reconstructions of Leishmania mexicana through the cell cycle” (PPATHOGENS-D-23-02228) for review by PLoS Pathogens. Your manuscript was evaluated at the editorial level and by three independent reviewers. These reviewers were strongly supportive of this manuscript but identified a few issues that should be addressed to improve the presentation. We therefore ask that you modify the manuscript according to the reviewers’ recommendations or respond in your subsequent cover letter with reasons why you believe specific aspects do not require modification of the manuscript.

Overall, while reviewers considered the body of work to be on the descriptive rather than mechanistic side, they also agreed that the contribution to understanding subcellular structure and cell division provides an important contribution to the cell biology of Leishmania parasites. The approach using sophisticated electron microscopic 3-dimensional reconstruction methods goes well beyond the resolution of current light microscopic analyses, and this study will have a significant impact for others working in this discipline. I agree with the reviewers and the authors that the advances in this manuscript are of high value despite the current lack of mechanistic understanding.

I am returning your manuscript with the three reviews. I ask that you pay attention to the following points raised by the reviewers.

• Reviewer 1. Please make clear how new flagella were distinguished from old flagella (Figs. 1B and 2C). Is the distribution (Fig. 1E) consistent with prior studies? Please provide quantification (Fig. 2) confirming that the NF always forms to the right of the OF.

• Reviewer 2. Please discuss any ideas you may have about how organelles maintain their positions and then move when cells progress toward cytokinesis. Please include the two references mentioned by this reviewer by Akiyoshi regarding MAD2 and Marcelo Santos Da Silva, 2013.

• Reviewer 3. Please address this reviewer’s specific comments regarding Figs. 1E, 5I, and 5G-H.

If all of relevant points are addressed, I hope to be able to make a final decision without sending the manuscript back for a second round of review.

In addition, when you are ready to resubmit, please provide the following:

1) A cover letter containing a detailed list of your responses to the review comments and a description of the changes you have made to the manuscript.

2) Two versions of the manuscript: one with either highlights or tracked changes denoting where the text has been changed; the other a clean version (uploaded as the manuscript file).

Sincerely,

Scott M Landfear

Guest Editor

PLOS Pathogens

Margaret Phillips

Section Editor

PLOS Pathogens

Kasturi Haldar

Editor-in-Chief

PLOS Pathogens

orcid.org/0000-0001-5065-158X

Michael Malim

Editor-in-Chief

PLOS Pathogens

orcid.org/0000-0002-7699-2064

Reviewer Comments (if any, and for reference):

Reviewer's Responses to Questions

**Part I - Summary**

Reviewer #1: In this study, Hair et al used volume electron microscopy approaches to produce high resolution three-dimensional reconstructions of seven cell cycle stages of Leishmania mexicana promastigotes. It is an impressive collection of qualitative and quantitative analyses which should be valuable for the trypanosomatid cell biology community. While most results are consistent with previous studies on L. mexicana or other trypanosomatids, they do yield much better resolution. There is some new information including the dynamics of new flagellum formation, contractile vacuole, and nuclear pores which could lead to mechanistic and hypothesis-driven research in the future.

Reviewer #2: The manuscript by Hair et al, uses state of the art scanning electron microscopy and 3D tomography reconstructions to analyze the organelle distribution and morphological changes of Leishmania mexicana promastigotes during cell division. The authors present a detail analysis of the patterns observed in asynchronous cell culture as well as promastigotes collected from the stomodeal valve of infected sand flies.

They elegantly show discrete stages during cell division and quantify the abundance and volume of organelles. Their data supports a highly coordinated control for organelle replication and segregation, and it has previously shown by other techniques with less resolution. The data collection and analysis is rigorous and presented in a clear way.

This work reinforces the idea of molecular mechanisms that tightly control the progression of the cell cycle in Leishmania, still to be elucidated. A question that I wish the authors would address is, based on their observations, how do they think the organelles maintaining their position and eventually moving when the cells change shape and progress toward cytokinesis. Have they observed association with cytosolic microtubules or other cytoskeletal elements? What triggers the duplication of organelles at such discrete time points? Incorporating some comments about these points in the Discussion would be a valuable addition to the manuscript, that, otherwise is outstanding.

Reviewer #3: In this work, Hair and colleagues have employed SBF-SEM and tomographic reconstructions to to study the mechanism of cell division in promastigote Leishmania. The high information content and resolution of their datasets have allowed them to establish a set of cell division intermediates that identify the order of assembly of many key cellular components, such as the flagellum, FP, secretory pathway, and DNA-containing compartments. This work has led to confirmation of the order of events that had previously been seen using light microscopy, such as the timing of new flagellum growth and copy number of the contractile vacuole and Golgi. It also provides tantalizing glimpses into previously unknown phenomena, such as the association of the ER with non-MtQ subpellicular microtubules and the depletion of nuclear pores from the nuclear bridge. The fact that the cleavage furrow appears to ingress from both sides of the cell body and produces indentations that persist after the completion of cytokinesis is also a very interesting discovery.

**Part II – Major Issues: Key Experiments Required for Acceptance**

Reviewer #1: The authors state that a total of 40 log phase promastigotes whose cell body and flagella lay within the imaged dataset were analyzed. I understand that flagella may be detached during processing but how often does that happen? In other words, how representative are those 40 cells amongst the seven stages?

Reviewer #2: (No Response)

Reviewer #3: Overall, the quality of the work in the manuscript is excellent. The collection of imaging provides is a valuable resource to the field and provides a series of observations that could be the foundation for subsequent work. I will say that several of the observations tend to lack detail or context- such as the cap structure at the end of the flagellum and the ER-microtubule association. The lack of perturbations, either chemical or via protein knockout, means that we are looking at a purely observational body of work. I think the scope of the work and degree of quantitation do mitigate this issue.

**Part III – Minor Issues: Editorial and Data Presentation Modifications**

Reviewer #1: The last sentence of the abstract “Our insights into the cell cycle mechanics of Leishmania will be invaluable for future molecular cell biology analyses of these important parasites.” Seems a bit too strong and self-serving and needs to be toned down.

Fig. 1B and Fig. 2C: It is not clear how new flagella (NF) are distinguished from the old flagella. Do NF not have PFR? If so, when is the PFR made for the NF? The information should be referenced if available.

Fig. 1E: no error bars were included. Is the distribution consistent with prior studies on promastigote cell cycle using other approaches (e.g., fluorescence microscopy)?

Line 148: The word “dividing” is misspelled.

Fig. 2: Some kind of quantitation is required to establish the conclusion that NF always forms to the right of OF.

Fig. 4. “Nasecent” seems to a misspelling of nascent.

Fig. 6E-H: the labeling for sand fly and cultured promastigotes seems to be flipped (the text states sand fly parasites are smaller than culture promastigotes).

Reviewer #2: Minor corrections/questions:

- Why did you decide to look at the cells in asynchronous culture vs synchronous? Analyzing synchronized cells could provide clearer evidence on when does the new flagellum assembly starts (lines 427-430).

- The basal body is not represented in the reconstructions, even though is a key structure for the formation of the new flagellum. Could you explain why? Is it due to a technical limitation?

- Lines 27-29: please, revise the grammar of the sentence "We showed that..."

- Replace Golgi Body in for Golgi, as all the figure labels indicate only Golgi.

- Line 148 replace (dividng) for (dividing)

- In line 435 you cite the work of Akiyoshi regarding the possible control of the cell cycle initiation by MAD2. This is a preprint from 2020 that has not been peer-reviewed published. Are there other references supporting the role of this or other proteins in initiating cell cycle in trypanosomatids? This is a pending question that needs further study. It would be good to add more supporting literature.

- The work by Marcelo Santos Da Silva, 2013 in Leishmania amazonensis is an important reference that should be mentioned in the manuscript (lines 436-439).

Reviewer #3: Fig 1E- hard to see 6/7 on very dark grey background in the graph. I would recommend altering the color scheme.

5I: this is a very interesting way to represent the subpellicular microtubules. While the seam in the first stage 1 cell is fairly clear, in the second cell it looks like there are couple other features that look similar to what is called the seam, along with regions in the Stage 2 cells. Is it possible to employ a statistical test to show that the MT length in the seam region is significantly different from the rest of the array?

In Fig 5G-H, is is possible to use these numbers to estimate if the switch from stage 1 to stage 2 requires the production of more tubulin by the cells, or is the total net quantity of tubulin the same, with the depolymerized tubulin being used to extend the remaining microtubules? There would be some caveats to the measurement either way, but it would be very interesting if the net amount of tubulin in Stage 1 vs Stage 2 was similar.

PLOS authors have the option to publish the peer review history of their article (what does this mean?). If published, this will include your full peer review and any attached files.

Reviewer #1: **Yes: **Kai Zhang

Reviewer #2: No

Reviewer #3: No

Figure Files:

Data Requirements:

Reproducibility:

References:

---

## [Editor Report · Decision Letter 1]

15 Feb 2024

Dear Dr. Sunter,

We are pleased to inform you that your manuscript 'Whole cell reconstructions of Leishmania mexicana through the cell cycle' has been provisionally accepted for publication in PLOS Pathogens.

Best regards,

Scott M Landfear

Guest Editor

PLOS Pathogens

Margaret Phillips

Section Editor

PLOS Pathogens

Michael Malim

Editor-in-Chief

PLOS Pathogens

orcid.org/0000-0002-7699-2064

The modifications the authors have made in the revised manuscript in response to reveiwers' comments have dealt appropriately with each comment. I believe the manuscript is now ready for acceptance.

One minor editorial point is that the word "annulus" should be used in this singular, rather than the plural "annuli" form, on line 241 of the revised manuscript.
---

## [Editor Report · Acceptance letter]

22 Feb 2024

Dear Dr Sunter,

We are delighted to inform you that your manuscript, "Whole cell reconstructions of *Leishmania mexicana* through the cell cycle," has been formally accepted for publication in PLOS Pathogens.

Best regards,

Michael Malim

Editor-in-Chief

PLOS Pathogens

orcid.org/0000-0002-7699-2064